# MRFNet: Multi-Receptive Field Network for Multivariate Time-Series Prediction

## Abstract

Time series forecasting is a critical topic in machine learning. Although existing deep learning methods have demonstrated outstanding performance and currently dominate this field, the latest state-of-the-art (SOTA) models are increasingly encountering the same limitations: the blockneck of performance. We believe this convergence is due to these models being based on the same mathematical foundations. To address this issue, we draw inspiration from the universal approximation theorem (UAT) and show that most commonly used deep learning models for time series forecasting are specific implementations of UAT. Based on UAT theory and the characteristics of time series data, we propose a new forecasting model called the **M**ulti-**R**eceptive **F**ield **N**etwork (MRFNet). This architecture integrates linear, sparse matrix, convolutional, and Fourier transform modules, resulting in an interpretable model with multiple receptive fields that can capture both global and local information. The MRFNet model has been tested extensively on several popular time series forecasting datasets and has achieved superior results.

## 1 Introduction

Time series forecasting is a critical problem with widespread applications in areas such as weather forecasting, traffic prediction, stock market analysis, and power consumption forecasting. To tackle this issue, various analytical and statistical methods have been developed. Early research relied on statistical measures like mean and variance to construct models, such as ARMA (McLeod and Li, 1983) and ARIMA (Ho and Xie, 1998; Zhang, 2003). These models have shown impressive accuracy and robustness but have limitations in generalization ability and multivariate time series forecasting.

With the advent of deep learning, a series of deep learning-based time series forecasting models emerged, outperforming traditional machine learning methods. The development of deep learning methods for time series forecasting has largely mirrored the evolution of deep learning models themselves, starting with fully connected neural networks (Kourentzes et al., 2014; Rahman et al., 2015; Torres et al., 2018), followed by recurrent neural networks (RNNs) (Bouktif et al., 2018; Lai et al., 2018; Bahdanau et al., 2014; Hewamalage et al., 2021), and subsequently, convolutional neural networks, Linear model (Zeng et al., 2023), and Transformer networks (Zhou et al., 2021; Wu et al., 2021; Zhou et al., 2022; Nie et al., 2022).

In summary, the three main types of models dominating the time series forecasting field today are linear models, convolutional models, and Transformer models. However, as deep learning continues to evolve, these models have begun to approach a performance bottleneck on widely used public time series forecasting datasets. We believe that the convergence of their performance is due to their shared mathematical foundations. To delve deeper into this issue, we require a fundamental mathematical interpretation of these models. The cornerstone of current deep learning theory is the UAT (Cybenko, 2007). Therefore, our goal is to study time series forecasting networks within the framework of UAT. However, UAT traditionally applies only to perceptron networks, while other forms of networks are more complex and seem to diverge from the theorem. Thus, to investigate complex linear, convolutional, and Transformer models within the UAT framework, we need to express them in a mathematical form consistent with UAT. In UAT2LLMs (Wang and Li, 2024a) and UAT2CVs (Wang and Li, 2024b), the "Matrix-Vector Method" was proposed and used to unify 2D and 3D convolutional, and Transformer-based models within the UAT theoretical framework.

Building on these studies, we aim to unify deep learning time series forecasting models within the UAT framework.

Through our derivations, we demonstrate that most deep learning models in the time series forecasting domain, whether used independently or as combinations of the three fundamental modules - Linear, Convolution, and Transformer - are essentially specific implementations of UAT. This explains why they ultimately converge towards similar performance blockneck, as they represent different manifestations of the same mathematical model.

Based on UAT theory and the characteristics of time series forecasting problems, we propose the MRFNet model, which integrates linear, 1D convolutional, sparse matrix transformation, and Fourier transform modules. We compare this model with SOTA methods on commonly used time series forecasting datasets and our model achieves SOTA performances on many datasets. Additionally, we performed data preprocessing tailored to the characteristics of time series forecasting data, further enhancing performance. In conclusion, this paper aims to bridge the theoretical gap and practices in time series forecasting. Our goal is to provide researchers and practitioners with tools to understand and advance the theory and practice of time series forecasting. Our main contributions include:

- We have demonstrated that most of the current deep-learning-based time series prediction models are specific implementations of the UAT, and explained why the current SOTA models converge to similar performance bottlenecks.

- We proposed the MRFNet model based on the characteristics of time series prediction, which balances both global and local temporal learning. Our model achieved SOTA performances across multiple datasets, and we also proved that this model is a specific implementation of UAT for time series prediction.

- We rearranged the data based on the inherent characteristics of time series data, which further improved the predictive performance of the model.

The structure of this paper is organized as follows: Section 2 provides an overview of existing time series forecasting models. In Section 3, we introduce the UAT and the Matrix-Vector Method and discuss their roles in unifying various foundational deep learning modules. In Section 4, we first demonstrate how 1D convolution can be expressed in matrix-vector form using this method, and then present the UAT formulation for general models in time series prediction. Section 5 introduces the MRFNet model (Section 5.1) and describes the time series datasets used (Section 5.2). Section 5.3 presents the prediction results of MRFNet and compares them with SOTA models. In Section 5.4, we apply feature selection tailored to the characteristics of time series data to further enhance the model's performance. Finally, in Section 5.5, we provide evidence that current time series forecasting models, being specific implementations of UAT, inevitably converge towards similar performance bottlenecks.

## 2 RELATED WORK

**RNN**: Recurrent Neural Networks (RNNs) were very popular for time series forecasting a few years ago, emphasizing the importance of sequence dependencies. RNNs incorporate various gating units to learn connections between different sequence positions (Bouktif et al., 2018; Lai et al., 2018; Bahdanau et al., 2014; Hewamalage et al., 2021). RNNs are fundamentally based on the Markov chain process in mathematics. However, key bottlenecks remain, such as vanishing gradients, substantial training workloads, and rapid error accumulation over longer time spans (Ribeiro et al., 2020).

**CNN**: The use of Convolutional Neural Networks (CNNs) for audio-related problems was initially proposed by Oord et al. (2016). Later, Bai et al. (2018) introduced the concept of Temporal Convolutional Networks (TCNs) as an alternative for time series forecasting (Bai et al., 2018; Vorbach et al., 2021; Aksan and Hilliges, 2019; Luo and Mesgarani, 2019; Hewage et al., 2020). This approach is influenced by the autoregressive Wavenet model (Oord et al., 2016) and incorporates causal convolutions to prevent the use of future information. Additionally, TCNs employ dilated convolutions to capture long-term dependencies in time series data. Several similar models have been developed, including works by Lara-Benítez et al. (2020); Wan et al. (2019); Liu et al. (2021).

**Transformer**: Transformers have dominated deep learning and shown significant potential in solving time series forecasting problems (Li et al., 2019; Wu et al., 2020; Lim et al., 2021; Wen et al., 2022). The multi-head attention architecture can extract information while positional embeddings help retain sequence position information (Kitaev et al., 2020; Zhang and Zhu, 2018; Wu et al., 2021; Shen and Wang, 2022; Madhusudhanan et al., 2021). However, Transformers are computationally complex, and the setting of hyperparameters greatly influences the performance of Transformer-based models. To address these issues, models like Informer, Autoformer, and Fedformer were developed (Zhou et al., 2021; Wu et al., 2021; Zhou et al., 2022). Liu et al. (2022) tackles time series problems from the perspective of stationarity, while ETSformer (Woo et al., 2022) uses exponential smoothing and frequency attention to replace the self-attention mechanism in Transformers, enhancing accuracy and efficiency. Additionally, Zhou et al. (2023) explored using pretrained large language models fine-tuned for time series prediction.

**Linear**: Zeng et al. (2023) argued that Transformers are not the optimal solution for time series forecasting. Instead, they demonstrated that linear methods could achieve SOTA results for both long-range and short-term time series forecasting, outperforming existing models.

## 3   THE UNIVERSAL APPROXIMATION THEORY

Our objective is to explore time series prediction problems within the framework of the UAT. To begin, we provide a basic introduction to this theorem, originally proposed by Cybenko (2007), which encompasses various conclusions and proofs. In this context, we specifically use the form of the UAT presented in Cybenko (2007) as an illustrative example.

Theorem 2 in Cybenko (2007) asserts that if $\sigma$ is any continuous sigmoidal function, then finite sums of the form:

$$G(\mathbf{x}) = \sum_{j=1}^{N} \alpha_j \sigma \left( \mathbf{W}_j^{\mathrm{T}} \mathbf{x} + \theta_j \right) \tag{1}$$

are dense in $C\left(\mathbf{I}_n\right)$. There are two case for Eq. (1) as follows:

- If $f(\mathbf{x}) \in \mathbb{R}^m$: $G(\mathbf{x}) \in \mathbb{R}^m$ , $\mathbf{x} \in \mathbb{R}^n$, $\alpha_i$, $\mathbf{W}_i \in \mathbb{R}^{(n,m)}$, and $\theta_i \in \mathbb{R}^m$
- If $f(\mathbf{x}) \in \mathbb{R}$: $G(\mathbf{x}) \in \mathbb{R}$ , $\mathbf{x} \in \mathbb{R}^n$, $\mathbf{W}_i \in \mathbb{R}^n$ , and $\alpha_i, \theta_i \in \mathbb{R}$

$\mathbf{x}$ is the input. For any $f \in C\left(\mathbf{I}_n\right)$ and $\varepsilon > 0$, there exists a sum $G(\mathbf{x})$ of the above form for which:

$$|G(\mathbf{x}) - f(\mathbf{x})| < \varepsilon \quad \text{for all} \quad \mathbf{x} \in \mathbf{I}_n. \tag{2}$$

This implies that, for a sufficiently large value of $N$, a single-layer neural network can approximate any continuous function over a closed interval. Furthermore, Hornik et al. (1989) extends this result, demonstrating that multi-layer feedforward networks also adhere to the UAT, capable of approximating arbitrary Borel measurable functions.

However, the approximate construction of this theorem is not directly applicable to convolutional and Transformer networks, as their basic operations-such as convolution and multi-head attention mechanisms-are difficult to express in the form of matrix-vector multiplication (The basic element in UAT is matrix-vector multiplication: $\alpha_j \sigma \left( \mathbf{W}_j^{\mathrm{T}} \mathbf{x} + \theta_j \right)$, where $\mathbf{W}_j^{\mathrm{T}} \mathbf{x}$ can be seen as various modules in networks). Nevertheless, most of the basic solutions for time series problems are based on multi-layer convolutional and Transformer neural networks. These networks involve a series of fundamental operations, such as 1D convolution, multi-head attention, and linear transformations (where the term "linear" differs from the linear transformation in Eq. (1); in Eq. (1), "linear" refers to $\mathbf{W}_j^{\mathrm{T}} \mathbf{x}$ which is a matrix multiplying a vector, while here it refers to a matrix multiplying another matrix). Therefore, if we can rewrite the mathematical expressions of these fundamental operations in the form of a matrix multiplying a vector and prove that they share the same mathematical structure as UAT, it would imply that all conclusions derived from UAT could be applied to multi-layer network architectures composed of these components.

In UAT2LLMs, the Matrix-Vector Method was already used to represent multi-head attention and linear operations in Transformers as a matrix multiplying a vector: $\mathbf{W}\mathbf{x}$. This paper extends the Matrix-Vector Method proposed in UAT2LLMs to prove that 1D convolution can also be expressed

in the form of matrix-vector multiplication. The concept behind the matrix-vector method is illustrated in Figure 1. The basic transformation in a network (e.g., 1D convolution) can be understood as $T(\mathbf{W})$, with inputs and outputs represented by $\mathbf{x}$ and $\mathbf{y}$, respectively. The Matrix-Vector Method transforms the input, output, and the parameters of the operation $T(\mathbf{W})$ within the network into $\mathbf{x}'$, $\mathbf{y}'$, and $\mathbf{W}'$, ensuring that $\mathbf{W}'\mathbf{x}' = \mathbf{y}'$. Note: For convenience, we use the symbol $'$ to denote the matrix-vector form of the corresponding variable. For example, $\mathbf{x}$ is represented as $\mathbf{x}'$.

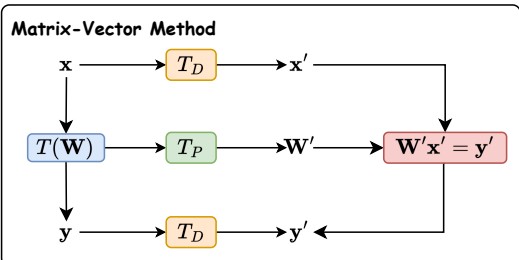

Figure 1: A general description to the Matrix-Vector Method.

# 4 UNIVERSAL APPROXIMATION THEORY FOR TIME SERIES PREDICTION

Currently, deep learning models for time series forecasting can be summarized as multi-layer networks composed primarily of one or more of the following modules: linear, convolution, and Transformer. These modules are typically connected using residual connections between successive layers. To demonstrate that such multi-layer networks, composed of these basic modules, are concrete implementations of the UAT, we first need to show that these modules can be represented in matrix-vector form. Once this representation is established, we then need to prove that the mathematical form of these networks with residual connections aligns with the UAT equation, Eq. (1).

In UAT2LLMs, it has already been shown that networks composed of multiple Transformer layers have a mathematical form consistent with UAT, and that linear operations can be expressed in matrix-vector form. Therefore, our primary focus here is to prove that 1D convolutional networks can also be represented in matrix-vector form (see Section 4.1). (Note that we will use a special calculation called diamond multiplication: $\diamond$, and $\mathbf{W} \diamond \mathbf{x} = \mathbf{W}^T\mathbf{x}$. More details could be found in UAT2LLMs) Following that, we demonstrate that the mathematical form of multi-layer networks with residual connections, expressed in matrix-vector terms, is consistent with UAT (see Section 4.2).

## 4.1 THE MATRIX-VECTOR FORMAT OF 1D CONVOLUTION

In this section, we will transform 1D convolution into the matrix-vector format. Figure 2.Single-channel Output.a vividly illustrates the basic process of single-channel output in 1D convolution: $\mathbf{W} \circledast \mathbf{x} = \mathbf{y}$. Here, the dimension $T$ symbolizes the time size, $K$ is the kernel size, while $N$ represents the feature size. Figure 2.Single-channel Output.b provides a concise example of single-channel output in 1D convolution. Figure 2.Single-channel Output.c further demonstrates how to use the matrix-vector method to transform the convolution of single-channel output into the form of diamond multiplication:$\mathbf{W}' \diamond \mathbf{x}' = \mathbf{y}'$, where $\mathbf{x}'$,$\mathbf{W}'$ and $\mathbf{y}'$ are generated from $\mathbf{x}$,$\mathbf{W}$ and $\mathbf{y}$.

Figure 2.Multi-channel Output.a elaborates on the fundamental process of 1D convolution with multiple convolution kernels and multiple-channel output:$\mathbf{W}_1 \circledast \mathbf{x} = \mathbf{y}_1...\mathbf{W}_O \circledast \mathbf{x} = \mathbf{y}_O$, where the number of output channels $O$ signifies predictions for the next $O$ time steps. Figure 2.Multi-channel Output.b offers a simplified example of Figure 2.Multi-channel Output.a, and Figure 2.Multi-channel Output.c employs the matrix-vector method to convert the convolution of multiple-channel output into the form of diamond multiplication:$\mathbf{W}' \diamond \mathbf{x}' = \mathbf{y}'$, where $\mathbf{W}'$ is generated from $\mathbf{W}_1...\mathbf{W}_O$ and $\mathbf{y}'$ is generated from $\mathbf{y}_1...\mathbf{y}_O$. With the relationship between diamond multiplication and matrix multiplication, we can derive the following formula:

$$\mathbf{x}_{i+1} = \mathbf{W}_i \circledast \mathbf{x}_i \rightarrow (\mathbf{x}^{i+1})' = \mathbf{W}'_i \diamond \mathbf{x}'_i = (\mathbf{W}'_i)^T\mathbf{x}'_i \tag{3}$$

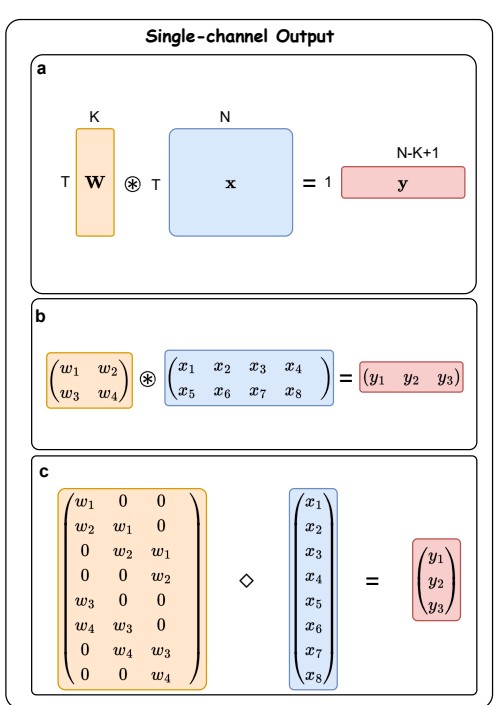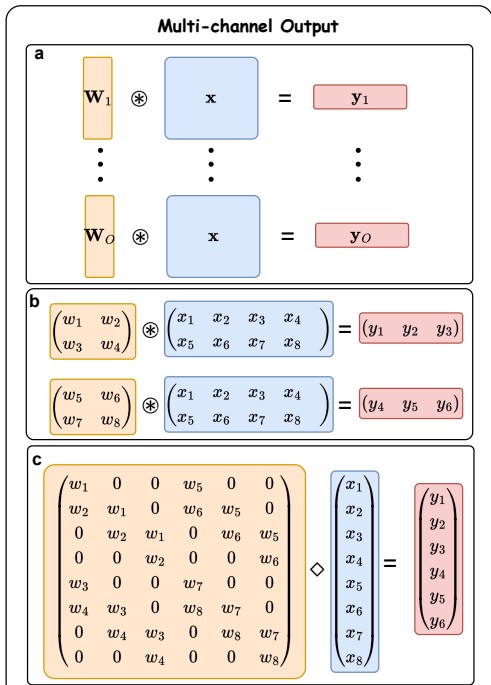

Figure 2: Illustration on how 1D convolution can be represented in matrix-vector form. The left side corresponds to a 1D convolution with a single output channel, while the right side represents a 1D convolution with multiple output channels. The convolution kernel, input data, and output data are indicated by orange, blue, and pink boxes, respectively. (a) Conceptual diagram of a 1D convolution, using different colored boxes to represent data and parameters. (b) A simple example of a 1D convolution. (c) The matrix-vector representation for the 1D convolution in (b).

Here, $\mathbf{x}_i \in \mathbb{R}^{(N,M)}$ represents the input of the $i$-th layer, while $\mathbf{x}_{i+1} \in \mathbb{R}^{(O,M-k+1))}$ is the output from the same layer. The matrix $\mathbf{W}_i \in \mathbb{R}^{(O,N,K)}$ are convolution kernel with the kernel size is $K$ and the number of kernel is $O$. $\mathbf{x}'_{i+1}$, $\mathbf{x}'_i$, $\mathbf{W}'_i$ are transformed from $\mathbf{x}_{i+1} \in \mathbb{R}^{(O(M-k+1),1)}$, $\mathbf{x}_i \in \mathbb{R}^{(NM,1)}$, $\mathbf{W}_i \in \mathbb{R}^{(NM,O(M-k+1)}$ based on matrix-vector method.

So we have proved the 1D convolution in time series can be represented in matrix-vector format. In the context of time series prediction, we observe that a single operation of 1D convolution learns the correlations among $K$ features at all time points using the convolution kernel. It generates output for a single feature at a specific time step. Sliding the convolution kernel can produce output for multiple features at the same time step, and increasing the number of convolution kernels enables obtaining outputs for multiple features across multiple time steps.

## 4.2 THE UAT FORMAT OF DEEP LEARNING NETWORKS IN TIME SERIES PREDICTION

Currently, the main deep learning networks used in time series forecasting are based on linear, convolutional and Transformer modules, which are typically connected in a residual form. So our purpose is to prove that the mathematical format of residual-based networks is consistent with UAT. It has already been proven in UAT2LLMs that the mathematical form of multi-layer Transformer networks is consistent with that of UAT, which can be written as:

$$\mathbf{x}_{i+1} = (\mathbf{W}'_{i+1,1}\mathbf{x}_0 + \mathbf{b}_{i+1,1}) + \sum_{j=1}^{i+1} \mathbf{W}'_{j,3}\sigma(\mathbf{W}'_{j,2}\mathbf{x}'_0 + \mathbf{b}'_{j,2}) \tag{4}$$

where $\mathbf{x}_{i+1}$ is the output of $i+1$-th layer, $\mathbf{x}_0$ is the input of the network, $\mathbf{b}'_{j,2} = (\mathbf{W}'_{j,2}\mathbf{b}'_{j-1,3} + \mathbf{b}'_{j,2}) + \mathbf{W}'_{j,2}UAT^R_{j-1}$, where $UAT^R_{j-1} = \sum_{k=1}^{j-1} \mathbf{W}'_{k,3}\sigma(\mathbf{W}'_{k,2}\mathbf{x}'_0 + \mathbf{b}'_{k,2})$. The term $\mathbf{b}'_{j,2}$ is approx-

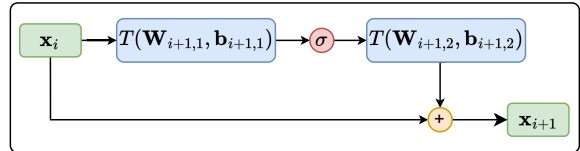

Figure 3: A general description of the residual-based module in a network.

imated by the $j$ layer of UAT with $\mathbf{x}_0$ as input. This enhances the model's ability to dynamically adjust functions based on input.

Here, we aim to prove that the mathematical form of residual-based multi-layer 1D convolutional and Linear networks is also consistent with UAT. Given the diversity of network architectures, we assume, without loss of generality, that the network structure is as depicted in Figure 3.

In this structure, $T$ represents either a linear or a convolution, while $\mathbf{W}_i$ and $\mathbf{b}_i$ denote the corresponding parameters. The mathematical representation of the network shown in Figure 3 can then be expressed as follows:

$$\mathbf{x}_{i+1} = \mathbf{x}_i + T_{i+1}\{\sigma[T_i(\mathbf{x}_i)]\} \tag{5}$$

Since we have demonstrated that 1D convolution can be expressed in the form of matrix-vector multiplication, and UAT2LLMs has shown that linear operations can also be expressed as matrix-vector multiplication, Eq. (5) can therefore be rewritten as:

$$\begin{aligned}\mathbf{x}'_{i+1} &= \mathbf{x}'_i + \mathbf{W}'_{i+1,2}\sigma(\mathbf{W}'_{i+1,1}\mathbf{x}'_i + \mathbf{b}'_{i+1,1}) + \mathbf{b}'_{i+1,2} \\ &= (\mathbf{x}'_i + \mathbf{b}'_{i+1,2}) + \mathbf{W}'_{i+1,2}\sigma(\mathbf{W}'_{i+1,1}\mathbf{x}'_i + \mathbf{b}'_{i+1,1})\end{aligned} \tag{6}$$

Eq. (6) is mathematically identical to the residual-based convolutional neural networks in UAT2CVs. UAT2CVs has already demonstrated that the mathematical form of a multi-layer network, as expressed in Eq. (6), is mathematically consistent with UAT, which is given by:

$$\mathbf{x}'_{i+1} = (\mathbf{x}'_0 + \mathbf{b}'_{i+1,2}) + \Sigma_{j=1}^{i+1}\mathbf{W}'_{j,2}\sigma(\mathbf{W}'_{j,1}\mathbf{x}'_0 + \mathbf{b}'_{j,1}) \tag{7}$$

where $\mathbf{b}'_{j,1} = (\mathbf{W}'_{j,1}\mathbf{b}'_{j-1,2} + \mathbf{b}'_{j,1}) + \mathbf{W}'_{j,1}UAT^R_{j-1}$ and $UAT^R_{j-1} = \Sigma_{k=1}^{j-1}\mathbf{W}'_{k,2}\sigma(\mathbf{W}'_{k,1}\mathbf{x}'_0 + \mathbf{b}'_{k,1})$. Eq. (7) conforming to the UAT form given in Eq. (1). In this equation, $\mathbf{b}_{j,1}$ is a dynamic parameter approximated by $j$ layers UAT ($j > 1$).

According to the above derivation, we know most deep-learning neural networks are the implementations of UAT.

## 5 MRFNET AND EXPERIMENTS

In this section, we first introduce the MRFNet architecture and discuss its key features (Section 5.1). Next, we describe the datasets used in our study (Section 5.2) and compare the performances of our model with several current SOTA models to demonstrate its effectiveness (Section 5.4). Following this, we leverage the characteristics of time series data to further enhance the model's performances (Section 5.4). Finally, in Section 5.5, we use experiments to validate that these UAT-based models ultimately converge to the same performance bottleneck.

### 5.1 MRFNET

Based on the characteristics of time series data, we designed the MRFNet model for time series forecasting. The architecture of MRFNet is illustrated in Figure 4: first, a linear module is used to embed time information. In time series forecasting, the changes in the sequence are often influenced by natural factors, such as seasonality and periodicity. Therefore, it is necessary to embed time

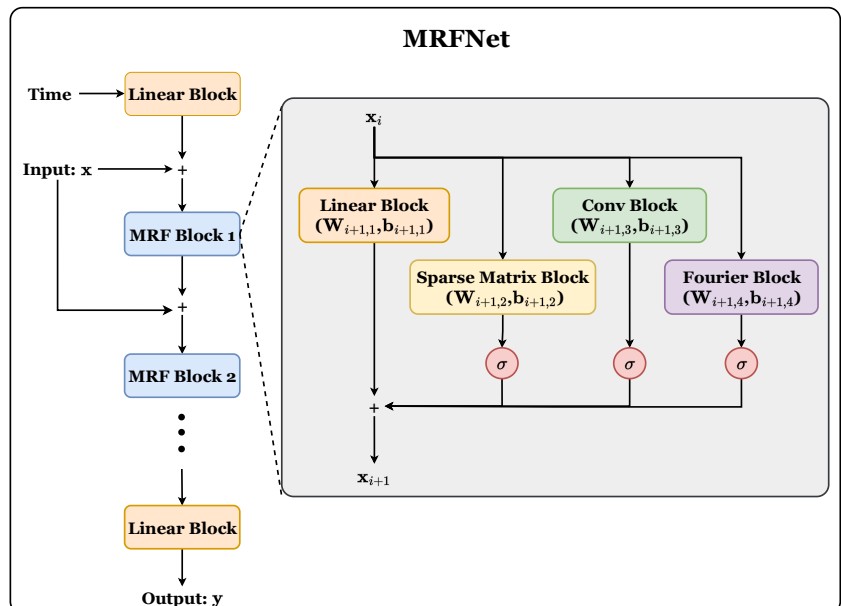

Figure 4: Overview of the MRFNet model's architecture. The details of the MRF block are shown on the right side.

information. The model then leverages multiple MRF modules for learning, and finally, a Linear block is applied to produce the output.

The MRF module consists of four main components: the Linear Block, Sparse Matrix Block, Conv Block, and Fourier Block. The Linear Block, which can be represented as $\mathbf{W}_{i+1,1}\mathbf{x} + \mathbf{b}_{i+1,1}$, performs linear transformations on the input data, capturing the overall temporal trends within the data. Given that features often share common attributes, we introduce a Linear Block to learn these shared characteristics, aiming to capture the intrinsic commonalities within the data. The Sparse Matrix Block focuses on learning local temporal features, which can be represented as $(\mathbf{W}_S \odot \mathbf{W}_{i+1,2})\mathbf{x} + \mathbf{b}_{i+1,2}$. Where $\mathbf{W}_S$ is a custom sparse matrix composed of 0s and 1s, ensuring the sparsity of $\mathbf{W}_{i+1,2}$.

The Conv Block employs 1D convolution, commonly used in time series forecasting, allowing the model to learn multi-feature correlations based on the size of the convolution kernel, represented as: $\mathbf{W}_{i+1,3} \circledast \mathbf{x}_i + \mathbf{b}_{i+1,3}$. The connection between convolution and the Fourier Block can be expressed as $\mathbf{x} \circledast \mathbf{h} = \mathbf{F}^{-1}[\mathbf{Fx} \odot \mathbf{Fh}]$. In the frequency domain, we have the capacity to learn $\mathbf{h}$, which is equivalent to $\mathbf{x} \circledast \mathbf{h} = \mathbf{F}^{-1}\mathbf{WFx}$. Thus, we introduce an enhanced convolutional variant in the neural network, $\mathbf{F}^{-1}\mathbf{WFx}$. This enables us to extract features in the frequency domain and achieve a broader receptive field, as demonstrated in the Appendix D. Additionally, the Fourier Transform is known for its ability to generate sparse representations, where only a few coefficients contain significant information, which can be used to emphasize key features.

Since we have proven that both linear transformations and convolutions can be expressed in a matrix-vector format, all four basic modules within the MRF Block can also be represented in this format. The Sparse Matrix Block can be considered a special type of linear transformation, while the Fourier Block can also be treated as a special case of linear transformation because $\mathbf{F}^{-1}$ and $\mathbf{F}$ are fixed. Therefore, we can define $\mathbf{W}_F = \mathbf{F}^{-1}\mathbf{WFx}$, which allows the Fourier Block to be viewed as a special Linear transformation. Consequently, the general form of MRFNet can be expressed as:

$$
\begin{aligned}
\mathbf{x}'_{i+1} = \mathbf{W}'_{i+1,1}\mathbf{x}'_i + \mathbf{b}'_{i+1,1} + \sigma(\mathbf{W}'_{i+1,2}\mathbf{x}'_i + \mathbf{b}'_{i+1,2}) \\
+ \sigma(\mathbf{W}'_{i+1,3}\mathbf{x}'_i + \mathbf{b}'_{i+1,3}) + \sigma(\mathbf{W}'_{i+1,4}\mathbf{x}'_i + \mathbf{b}'_{i+1,4})
\end{aligned}
\tag{8}
$$

Based on Eq. (8), it is easy to prove that MRFNet is also an implementation of UAT and effectively increases the depth of the UAT. We provide the proof in Appendix B. The various transformations

Table 1: A general description of all datasets.

| Datasets | ETTh1&ETTh2 | ETTm1 &ETTm2 | Traffic | Electricity | Weather | ILI |
|---|---|---|---|---|---|---|
| Variates | 7 | 7 | 862 | 321 | 21 | 7 |
| Timesteps | 17,420 | 69,680 | 17,544 | 26,304 | 52,696 | 966 |
| Granularity | 1hour | 5min | 1hour | 1hour | 10min | 1week |

Table 2: Multivariate predictions of ETTh1, ETTh2, ETTm1, ETTm2, Traffic, Electricity, Weather and ILI, by twelve models. The best results are highlighted in bold red. The second-best results are indicated with highlighted in bold black. Here, we provide a comparison of only some models. A more comprehensive comparison is presented in Table 5 in the Appendix.

| Methods | | MRFNet | | GPT2(6) | | DLinear | | PatchTST | | TimesNet | | FEDformer | | Autoformer | |
|---|---|---|---|---|---|---|---|---|---|---|---|---|---|---|---|---|
| Metric | | MSE | MAE | MSE | MAE | MSE | MAE | MSE | MAE | MSE | MAE | MSE | MAE | MSE | MAE |
| Weather | 96 | 0.149 | 0.191 | 0.162 | 0.212 | 0.176 | 0.237 | 0.149 | 0.198 | 0.172 | 0.220 | 0.217 | 0.296 | 0.266 | 0.336 |
| | 192 | 0.193 | 0.235 | 0.204 | 0.248 | 0.220 | 0.282 | 0.194 | 0.241 | 0.219 | 0.261 | 0.276 | 0.336 | 0.307 | 0.367 |
| | 336 | 0.245 | 0.278 | 0.254 | 0.286 | 0.265 | 0.319 | 0.245 | 0.282 | 0.280 | 0.306 | 0.339 | 0.380 | 0.359 | 0.395 |
| | 720 | 0.315 | 0.329 | 0.326 | 0.337 | 0.333 | 0.362 | 0.314 | 0.334 | 0.365 | 0.359 | 0.403 | 0.428 | 0.419 | 0.428 |
| | Avg | **0.225** | **0.258** | 0.237 | 0.270 | 0.248 | 0.300 | **0.225** | 0.264 | 0.259 | 0.287 | 0.309 | 0.360 | 0.338 | 0.382 |
| ETTh1 | 96 | 0.364 | 0.393 | 0.376 | 0.397 | 0.375 | 0.399 | 0.370 | 0.399 | 0.384 | 0.402 | 0.376 | 0.419 | 0.449 | 0.459 |
| | 192 | 0.402 | 0.415 | 0.416 | 0.418 | 0.405 | 0.416 | 0.413 | 0.421 | 0.436 | 0.429 | 0.420 | 0.448 | 0.500 | 0.482 |
| | 336 | 0.442 | 0.444 | 0.442 | 0.433 | 0.439 | 0.443 | 0.422 | 0.436 | 0.491 | 0.469 | 0.459 | 0.465 | 0.521 | 0.496 |
| | 720 | 0.434 | 0.454 | 0.477 | 0.456 | 0.472 | 0.490 | 0.447 | 0.466 | 0.521 | 0.500 | 0.506 | 0.507 | 0.514 | 0.512 |
| | Avg | **0.410** | **0.426** | 0.427 | **0.426** | 0.422 | 0.437 | **0.413** | 0.430 | 0.458 | 0.450 | 0.440 | 0.460 | 0.496 | 0.487 |
| ETTh2 | 96 | 0.273 | 0.330 | 0.285 | 0.342 | 0.289 | 0.353 | 0.274 | 0.336 | 0.340 | 0.374 | 0.358 | 0.397 | 0.346 | 0.388 |
| | 192 | 0.341 | 0.376 | 0.354 | 0.389 | 0.383 | 0.418 | 0.339 | 0.379 | 0.402 | 0.414 | 0.429 | 0.439 | 0.456 | 0.452 |
| | 336 | 0.366 | 0.396 | 0.373 | 0.407 | 0.448 | 0.465 | 0.329 | 0.380 | 0.452 | 0.452 | 0.496 | 0.487 | 0.482 | 0.486 |
| | 720 | 0.385 | 0.423 | 0.406 | 0.441 | 0.605 | 0.551 | 0.379 | 0.422 | 0.462 | 0.468 | 0.463 | 0.474 | 0.515 | 0.511 |
| | Avg | **0.341** | **0.381** | 0.354 | 0.394 | 0.431 | 0.446 | **0.330** | **0.379** | 0.414 | 0.427 | 0.437 | 0.449 | 0.450 | 0.459 |
| ETTm1 | 96 | 0.297 | 0.342 | 0.292 | 0.346 | 0.299 | 0.343 | 0.290 | 0.342 | 0.338 | 0.375 | 0.379 | 0.419 | 0.505 | 0.475 |
| | 192 | 0.334 | 0.366 | 0.332 | 0.372 | 0.335 | 0.365 | 0.332 | 0.369 | 0.374 | 0.387 | 0.426 | 0.441 | 0.553 | 0.496 |
| | 336 | 0.366 | 0.385 | 0.366 | 0.394 | 0.369 | 0.386 | 0.366 | 0.392 | 0.410 | 0.411 | 0.445 | 0.459 | 0.621 | 0.537 |
| | 720 | 0.407 | 0.411 | 0.417 | 0.421 | 0.425 | 0.421 | 0.416 | 0.420 | 0.478 | 0.450 | 0.543 | 0.490 | 0.671 | 0.561 |
| | Avg | **0.351** | **0.376** | 0.352 | 0.383 | 0.357 | **0.378** | **0.351** | 0.380 | 0.400 | 0.406 | 0.448 | 0.452 | 0.588 | 0.517 |
| ETTm2 | 96 | 0.163 | 0.246 | 0.173 | 0.262 | 0.167 | 0.269 | 0.165 | 0.255 | 0.187 | 0.267 | 0.203 | 0.287 | 0.255 | 0.339 |
| | 192 | 0.219 | 0.287 | 0.229 | 0.301 | 0.224 | 0.303 | 0.220 | 0.292 | 0.249 | 0.309 | 0.269 | 0.328 | 0.281 | 0.340 |
| | 336 | 0.275 | 0.323 | 0.286 | 0.341 | 0.281 | 0.342 | 0.274 | 0.329 | 0.321 | 0.351 | 0.325 | 0.366 | 0.339 | 0.372 |
| | 720 | 0.354 | 0.377 | 0.378 | 0.401 | 0.397 | 0.421 | 0.362 | 0.385 | 0.408 | 0.403 | 0.421 | 0.415 | 0.433 | 0.432 |
| | Avg | **0.252** | **0.333** | 0.266 | 0.326 | 0.267 | 0.333 | **0.255** | **0.315** | 0.291 | 0.333 | 0.305 | 0.349 | 0.327 | 0.371 |
| ILI | 24 | 1.757 | 0.857 | 2.063 | 0.881 | 2.215 | 1.081 | 1.319 | 0.754 | 2.317 | 0.934 | 3.228 | 1.260 | 3.483 | 1.287 |
| | 36 | 2.085 | 0.915 | 1.868 | 0.892 | 1.963 | 0.963 | 1.430 | 0.834 | 1.972 | 0.920 | 2.679 | 1.080 | 3.103 | 1.148 |
| | 48 | 1.972 | 0.899 | 1.790 | 0.884 | 2.130 | 1.024 | 1.553 | 0.815 | 2.238 | 0.940 | 2.622 | 1.078 | 2.669 | 1.085 |
| | 60 | 1.998 | 0.923 | 1.979 | 0.957 | 2.368 | 1.096 | 1.470 | 0.788 | 2.027 | 0.928 | 2.857 | 1.157 | 2.770 | 1.125 |
| | Avg | 1.953 | **0.898** | **1.925** | 0.903 | 2.169 | 1.041 | **1.443** | **0.797** | 2.139 | 0.931 | 2.847 | 1.144 | 3.006 | 1.161 |
| ECL | 96 | 0.127 | 0.218 | 0.139 | 0.238 | 0.140 | 0.237 | 0.129 | 0.222 | 0.168 | 0.272 | 0.193 | 0.308 | 0.201 | 0.317 |
| | 192 | 0.144 | 0.234 | 0.153 | 0.251 | 0.153 | 0.249 | 0.157 | 0.240 | 0.184 | 0.289 | 0.201 | 0.315 | 0.222 | 0.334 |
| | 336 | 0.159 | 0.251 | 0.169 | 0.266 | 0.169 | 0.267 | 0.163 | 0.259 | 0.198 | 0.300 | 0.214 | 0.329 | 0.231 | 0.338 |
| | 720 | 0.192 | 0.280 | 0.206 | 0.297 | 0.203 | 0.301 | 0.197 | 0.290 | 0.220 | 0.320 | 0.246 | 0.355 | 0.254 | 0.361 |
| | Avg | **0.155** | **0.245** | 0.167 | 0.263 | 0.166 | 0.263 | **0.161** | 0.252 | 0.192 | 0.295 | 0.214 | 0.327 | 0.227 | 0.338 |
| Traffic | 96 | 0.386 | 0.241 | 0.388 | 0.282 | 0.410 | 0.282 | 0.360 | 0.249 | 0.593 | 0.321 | 0.587 | 0.366 | 0.613 | 0.388 |
| | 192 | 0.399 | 0.248 | 0.407 | 0.290 | 0.423 | 0.287 | 0.379 | 0.256 | 0.617 | 0.336 | 0.604 | 0.373 | 0.616 | 0.382 |
| | 336 | 0.414 | 0.274 | 0.412 | 0.294 | 0.436 | 0.296 | 0.392 | 0.264 | 0.629 | 0.336 | 0.621 | 0.383 | 0.622 | 0.337 |
| | 720 | 0.457 | 0.277 | 0.450 | 0.312 | 0.466 | 0.315 | 0.432 | 0.286 | 0.640 | 0.350 | 0.626 | 0.382 | 0.660 | 0.408 |
| | Avg | **0.414** | **0.260** | **0.414** | 0.294 | 0.433 | 0.295 | **0.390** | **0.263** | 0.620 | 0.336 | 0.610 | 0.376 | 0.628 | 0.379 |

within the MRF module can all be represented as corresponding matrix-vector multiplications, differing only in the size of the learned receptive field. By efficiently handling different temporal components and adjusting the receptive field, the model can capture subtle variations in the data.

## 5.2 DATASETS

The experiments were conducted on real-world datasets (Zhou et al., 2021), including (1) Electricity Transformer Temperature (ETT), (2) Electricity, and (3) Traffic, (4) Weather, (5) ILI. The details of all datasets can be found in (Wu et al., 2021). The data source is available at github[1]. It should be noted that ETT consists of four different datasets: ETTh1, ETTh2, ETTm1, and ETTm2, each of which has seven variables. We evaluate our model using Mean Absolute Errors (MAE) and Mean Squared Errors (MSE), as used in (Zhou et al., 2021). Smaller values of MAE/MSE indicate better

---

[1]https://github.com/thuml/Autoformer

model performance. We use the average values for all predictions. The details of all datasets are shown in Table 1. In Appendix A, we give the data setting for training, evaluation and testing.

## 5.3 RESULTS OF MRFNET

In Table 2, we compare our model with the current SOTA models for time series prediction. Our model demonstrates superior performances on these datasets, achieving SOTA results in most cases (except for the ILI dataset, which is primarily affected by the characteristics of the data itself; we will provide a solution for this issue in Section 5.4). However, when analyzing the results across all models, it becomes evident that the current SOTA models - MRFNet, GPT2(6) (Zhou et al., 2023), DLinear (Zeng et al., 2023), and PatchTST (Nie et al., 2022) - converge towards a similar performance bottleneck. This is because they are all composed of multi-layer Transformers and Linear components, which we have proven to be specific implementations of the UAT, leading them to the same blockneck with limited data. Additional results for univariate prediction can be found in Appendix F.

## 5.4 DATA EFFECTS

A crucial characteristic of time series data is its periodicity. However, this periodicity may change over time due to external factors (e.g., advancements in science leading to a gradual increase in electricity demand). This gives data a specific property: data points closer to the prediction point have a greater impact on the results (e.g., when forecasting electricity demand for 2024, the data from 2023 is more relevant, while earlier years are less so).

Leveraging this property, we designated the beginning part of the ETT data as the validation set, kept the test set unchanged, and used the remaining data for training. The results, as shown in Table 3 for MRFNet*. As seen in Table 3, the results improved substantially after adjusting the allocation of the training and validation sets.

Another issue is the relatively poor performance of MRFNet on the ILI dataset. We believe this is primarily due to the nature of the data itself. Not all features in the ILI dataset are correlated (see Appendix C) and forcing all data to be trained together may lead to worse results. Based on the characteristics of the ILI data, we divided it into two groups and trained them separately. The results are shown in Table 4. For more details, please refer to the Appendix C.

Table 3: We re-divided the training and validation sets for the ETTh1, ETTh2, ETTm1, and ETTm2 datasets, then retrained the models. The test results were obtained on the same test sets.

| Models | | MRFNet | | MRFNet* | | PatchTST/64 | |
|---|---|---|---|---|---|---|---|
| Metric | | MSE | MAE | MSE | MAE | MSE | MAE |
| ETTh1 | 96 | **0.364** | **0.393** | **0.302** | **0.368** | 0.370 | 0.400 |
| | 192 | **0.402** | **0.415** | **0.343** | **0.397** | 0.413 | 0.429 |
| | 336 | 0.442 | 0.444 | **0.366** | **0.415** | **0.422** | **0.440** |
| | 720 | **0.434** | **0.454** | **0.387** | **0.446** | 0.447 | 0.468 |
| ETTh2 | 96 | **0.273** | **0.330** | **0.246** | **0.319** | 0.274 | 0.337 |
| | 192 | **0.341** | **0.376** | **0.308** | **0.359** | **0.341** | 0.382 |
| | 336 | 0.366 | 0.396 | **0.325** | **0.381** | 0.329 | **0.384** |
| | 720 | 0.385 | 0.423 | **0.353** | **0.408** | 0.379 | 0.422 |
| ETTm1 | 96 | 0.297 | 0.342 | **0.243** | **0.316** | 0.293 | 0.346 |
| | 192 | 0.334 | 0.366 | **0.280** | **0.342** | 0.333 | 0.370 |
| | 336 | **0.360** | **0.385** | **0.299** | **0.359** | 0.369 | 0.392 |
| | 720 | **0.407** | **0.411** | **0.353** | **0.396** | 0.416 | 0.420 |
| ETTm2 | 96 | **0.163** | **0.246** | **0.144** | **0.233** | 0.166 | 0.256 |
| | 192 | **0.219** | **0.287** | **0.197** | **0.273** | 0.223 | 0.296 |
| | 336 | 0.275 | 0.323 | **0.245** | **0.309** | **0.274** | 0.329 |
| | 720 | **0.354** | **0.377** | **0.306** | **0.357** | 0.362 | 0.385 |

## 5.5 ABLATION EXPERIMENT

We decompose the MRFNet model into three variants: LS (Linear-Sparse Matrix), LC (Linear-Convolution), and LF (Linear-Fourier Transform), and compare them against the full MRFNet

Table 4: We split the 7 features of the ILI dataset into two groups for separate training and prediction. The seven features are labeled as Class 1 to Class 7. Group 1: Class 1, 2, 3, 4, 5; Group 2: Class 6,7. Ori: Results from training with all features together. Split: Results from training after splitting the features into two groups.

| Methods | | Class 1 | | Class 2 | | Class 3 | | Class 4 | | Class 5 | | Class 6 | | Class 7 | |
|---|---|---|---|---|---|---|---|---|---|---|---|---|---|---|---|
| Metric | | MSE | MAE | MSE | MAE | MSE | MAE | MSE | MAE | MSE | MAE | MSE | MAE | MSE | MAE |
| Ori | Avg | **0.581** | **0.860** | **0.630** | **1.046** | **1.139** | **2.821** | **1.020** | **2.807** | **1.269** | **4.236** | **0.744** | **0.800** | **0.904** | **1.097** |
| Split | Avg | 0.366 | 0.233 | 0.339 | 0.220 | 0.294 | 0.160 | 0.304 | 0.199 | 0.367 | 0.254 | 0.188 | 0.103 | 0.184 | 0.094 |

model. Table 5 presents the results of our ablation study. The results indicate that while certain models perform less well on specific datasets, such as the LC model on the Weather dataset, their overall performance remains very close. Notably, the MRFNet model consistently ranks either best or second-best across all datasets.

Table 5: Ablation Experiment on of ETTh1, ETTm2, and Weather. The best results are highlighted in bold red. The second-best results are indicated with an underlined orange line. The best results are highlighted in bold red. The second-best results are indicated with an underlined orange line

| Models | | MRFNet | | LS | | LC | | LF | |
|---|---|---|---|---|---|---|---|---|---|
| Metric | | MSE | MAE | MSE | MAE | MSE | MAE | MSE | MAE |
| ETTm1 | 96 | **0.149** | **0.191** | 0.170 | 0.208 | 0.169 | 0.208 | **0.148** | **0.189** |
| | 192 | **0.193** | **0.235** | 0.211 | 0.249 | 0.212 | 0.248 | **0.193** | **0.235** |
| | 336 | **0.245** | **0.278** | 0.257 | 0.286 | 0.257 | 0.286 | **0.244** | **0.277** |
| | 720 | **0.315** | **0.329** | 0.321 | 0.335 | 0.322 | 0.336 | **0.312** | **0.326** |
| ETTm1 | 96 | 0.297 | **0.342** | 0.297 | **0.341** | **0.275** | 0.343 | **0.296** | **0.341** |
| | 192 | **0.334** | **0.366** | 0.337 | 0.365 | **0.335** | **0.365** | 0.336 | 0.368 |
| | 336 | **0.360** | **0.385** | 0.370 | 0.388 | 0.369 | 0.386 | **0.357** | **0.383** |
| | 720 | **0.407** | **0.411** | 0.421 | 0.415 | 0.425 | 0.421 | 0.409 | 0.413 |
| ETTm2 | 96 | 0.163 | **0.246** | **0.160** | **0.245** | **0.161** | 0.246 | **0.161** | **0.245** |
| | 192 | **0.219** | **0.287** | **0.215** | **0.284** | **0.215** | **0.284** | 0.221 | 0.286 |
| | 336 | 0.275 | **0.323** | **0.267** | **0.321** | **0.268** | 0.330 | 0.277 | 0.324 |
| | 720 | **0.354** | **0.377** | 0.354 | **0.375** | **0.352** | **0.375** | 0.362 | 0.380 |

These results support our conclusions: first, we have demonstrated that these models inherently share the same mathematical properties, which explains the similarity in their results. Second, MRFNet is equivalent to an increased network depth compared to the other models and benefits from its more comprehensive feature learning. As a result, MRFNet maintains SOTA performance or near-SOTA performance across all datasets, further validating its robustness and superiority.

Additionally, in Appendix E, we compare the weights of MRFNet, LS, LC, and LF at the same layer. The weights for the same dataset exhibit similar textures, suggesting that, in general, they are learning similar information patterns. Given that we have proven these models share the same mathematical framework, the primary differences lie in the size of their receptive fields, with each model's module playing a similar role. However, due to the different receptive field sizes, the specific patterns they learn may vary slightly, though the overall trend remains consistent.

## 6 CONCLUSION

In this paper, we demonstrate that most models in the field of time series forecasting are specific implementations of the UAT, which explains why current SOTA models in time series forecasting tend to converge towards the same performance bottleneck. Based on the principles of UAT and the characteristics of time series data, we have designed a new model, MRFNet, for time series prediction. MRFNet integrates linear modules, sparse matrix modules, convolutional modules, and Fourier transform modules, effectively capturing both global and local receptive field information. Through extensive testing on various common datasets, the MRFNet model has demonstrated its superiority, achieving SOTA-level performance. Additionally, by leveraging the intrinsic properties of time series data, we further refined the data, significantly enhancing the performance on certain datasets. Finally, we conducted experiments to confirm that time series forecasting models based on UAT theory eventually converge to a similar performance bottleneck.

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

## A  DATA ARRANGEMENT

In this section, we present the data arrangement for the train set, validation set, and test set. We give the default arrangement in Table 1. In Table 2, we describe data rearrangement about ETTh1&2 and ETTm1&2 in Section 5.5. Note: n represents the total number of time steps and T is 30*24.

Table 1: The default arrangement of data split.

| Dataset | Train Set | Val Set | Test Set |
|---------|-----------|---------|----------|
| Weather | D[:n*0.7] | D[n*0.7:n*0.8] | D[n*0.8:] |
| ETTh1&2 | D[:T*12] | D[T*12:T*16] | D[T*16:T*20] |
| ETTm1&2 | D[:T*12*4] | D[T*12*4:T*16*4] | D[T*16*4:T*20*4] |
| ILI | D[:n*0.6] | D[n*0.6:n*0.8] | D[n*0.8:] |
| ECL | D[:n*0.7] | D[n*0.7:n*0.8] | D[n*0.8:] |
| Traffic | D[:n*0.7] | D[n*0.7:n*0.8] | D[n*0.8:] |

Table 2: The data rearrangement of ETTh1&2 and ETTm1&2.

| Dataset | Train Set | Val Set | Test Set |
|---------|-----------|---------|----------|
| ETTh1&2 | D[T*2:T*16] | D[:T*2] | D[T*16:T*20] |
| ETTm1&2 | D[T*1*4:T*16*4] | D[:T*1*4] | D[T*16*4:T*20*4] |

## B  THE UAT FORMAT OF MRFNET

Here, we follow the proof process from UAT2LLMs and UAT2CVs to demonstrate that MRFNet is also a specific implementation of the UAT. We know the general format of MRFNet is:

$$
\begin{aligned}
\mathbf{x}'_{i+1} = \mathbf{W}'_{i+1,1}\mathbf{x}'_i + \mathbf{b}'_{i+1,1} + \sigma(\mathbf{W}'_{i+1,2}\mathbf{x}'_i + \mathbf{b}'_{i+1,2}) \\
+ \sigma(\mathbf{W}'_{i+1,3}\mathbf{x}'_i + \mathbf{b}'_{i+1,3}) + \sigma(\mathbf{W}'_{i+1,4}\mathbf{x}'_i + \mathbf{b}'_{i+1,4})
\end{aligned}
\tag{1}
$$

According to UAT2LLMs and UAT2CVs, the general term of MRFNet is similar to the general term in UAT2LLMs(C.2) and UAT2CVs (A.2) by only two additional terms. According to these two UAT2LLMs and UAT2CVs, if the mathematical format of $\mathbf{x}'_i$ is the same to UAT, then the mathematical format of $\mathbf{W}'_{i+1,1}\mathbf{x}'_i$ is also the same with UAT and $\sigma(\mathbf{W}'_{i+1,2}\mathbf{x}'_i + \mathbf{b}'_{i+1,2})$, $\sigma(\mathbf{W}'_{i+1,3}\mathbf{x}'_i + \mathbf{b}'_{i+1,3})$ and $\sigma(\mathbf{W}'_{i+1,4}\mathbf{x}'_i + \mathbf{b}'_{i+1,4})$ can be seen as three terms in UAT. So it is straightforward to deduce that if the mathematical form of $\mathbf{x}'_i$ in MRFNet is consistent with the UAT form, then the mathematical form of $\mathbf{x}'_{i+1}$ will also be consistent with the UAT form. It is also easy to derive that the mathematical forms of $\mathbf{x}'_1$ and $\mathbf{x}'_2$ in MRFNet are consistent with the UAT. Thus, by mathematical induction, the mathematical form of MRFNet aligns with the UAT.

For the clarity, we give the mathematical format of $\mathbf{x}'_1$ and $\mathbf{x}'_2$, let the input to MRF Block 1 be $\mathbf{x}_0$. Then, the matrix-vector form of $\mathbf{x}_1$ can be expressed as shown in Eq. (2). Here, $\mathbf{x}'_1$ can be considered as being approximated by a UAT with $N$ equal to 4.

$$
\begin{aligned}
\mathbf{x}'_1 = \mathbf{W}'_{1,1}\mathbf{x}'_0 + \mathbf{b}'_{1,1} + \sigma(\mathbf{W}'_{1,2}\mathbf{x}'_0 + \mathbf{b}'_{1,2}) \\
+ \sigma(\mathbf{W}'_{1,3}\mathbf{x}'_0 + \mathbf{b}'_{1,3}) + \sigma(\mathbf{W}'_{1,4}\mathbf{x}'_0 + \mathbf{b}'_{1,4})
\end{aligned}
\tag{2}
$$

Similarly, based on Eq. (2), we can derive $\mathbf{x}'_2$ as shown in Eq. (3). Define the following: $\mathbf{W}'_{2,1} = \mathbf{W}'_{2,1}\mathbf{W}'_{1,1}$, $\mathbf{b}'_{2,1} = \mathbf{W}'_{2,1}\mathbf{b}'_{1,1} + \mathbf{b}'_{2,1}$, $\mathbf{W}'_{2,2} = \mathbf{W}'_{2,2}\mathbf{W}'_{1,1}$, $\mathbf{W}'_{2,3} = \mathbf{W}'_{2,3}\mathbf{W}'_{1,1}$, $\mathbf{W}'_{2,4} = \mathbf{W}'_{2,4}\mathbf{W}'_{1,1}$, $\mathbf{b}'_{2,2} = (\mathbf{W}'_{2,2}\mathbf{b}'_{1,1} + \mathbf{b}'_{2,2}) + \mathbf{W}'_{2,2}\sigma(\mathbf{W}'_{1,2}\mathbf{x}'_0 + \mathbf{b}'_{1,2}) + \mathbf{W}'_{2,2}\sigma(\mathbf{W}'_{1,3}\mathbf{x}'_0 + \mathbf{b}'_{1,3}) + \mathbf{W}'_{2,2}\sigma(\mathbf{W}'_{1,4}\mathbf{x}'_0 + \mathbf{b}'_{1,4})$, $\mathbf{b}'_{2,3} = (\mathbf{W}'_{2,3}\mathbf{b}'_{1,1} + \mathbf{b}'_{2,3}) + \mathbf{W}'_{2,3}\sigma(\mathbf{W}'_{1,2}\mathbf{x}'_0 + \mathbf{b}'_{1,2}) + \mathbf{W}'_{2,3}\sigma(\mathbf{W}'_{1,3}\mathbf{x}'_0 + \mathbf{b}'_{1,3}) + \mathbf{W}'_{2,3}\sigma(\mathbf{W}'_{1,4}\mathbf{x}'_0 + \mathbf{b}'_{1,4})$, and $\mathbf{b}'_{2,4} = (\mathbf{W}'_{2,4}\mathbf{b}'_{1,1} + \mathbf{b}'_{2,4}) + \mathbf{W}'_{2,4}\sigma(\mathbf{W}'_{1,2}\mathbf{x}'_0 + \mathbf{b}'_{1,2}) + \mathbf{W}'_{2,4}\sigma(\mathbf{W}'_{1,3}\mathbf{x}'_0 + \mathbf{b}'_{1,3}) + \mathbf{W}'_{2,4}\sigma(\mathbf{W}'_{1,4}\mathbf{x}'_0 + \mathbf{b}'_{1,4})$. Therefore, $\mathbf{x}'_2$ can be expressed as shown in Eq. (4), which can be seen as 7 layers' UAT. Thus, an MRFNet with $i$ layers is equivalent to a UAT

with $3i+1$ layers, effectively increasing the number of layers in the UAT. This is because, according to UAT2LLMs and UAT2CVs, a network with $i$ layers generally corresponds to a UAT with $i+1$ layers.

$$
\begin{aligned}
\mathbf{x}'_2 = {} & \mathbf{W}'_{2,1}\mathbf{x}'_1 + \mathbf{b}'_{2,1} + \sigma(\mathbf{W}'_{2,2}\mathbf{x}'_1 + \mathbf{b}'_{1,2}) \\
& + \sigma(\mathbf{W}'_{2,3}\mathbf{x}'_1 + \mathbf{b}'_{2,3}) + \sigma(\mathbf{W}'_{2,4}\mathbf{x}'_1 + \mathbf{b}'_{2,4}) \\
= {} & \mathbf{W}'_{2,1}[\mathbf{W}'_{1,1}\mathbf{x}'_0 + \mathbf{b}'_{1,1} + \sigma(\mathbf{W}'_{1,2}\mathbf{x}'_0 + \mathbf{b}'_{1,2}) + \sigma(\mathbf{W}'_{1,3}\mathbf{x}'_0 + \mathbf{b}'_{1,3}) \\
& + \sigma(\mathbf{W}'_{1,4}\mathbf{x}'_0 + \mathbf{b}'_{1,4})] + \mathbf{b}'_{2,1} \\
& + \sigma\{\mathbf{W}'_{2,2}[\mathbf{W}'_{1,1}\mathbf{x}'_0 + \mathbf{b}'_{1,1} + \sigma(\mathbf{W}'_{1,2}\mathbf{x}'_0 + \mathbf{b}'_{1,2}) + \sigma(\mathbf{W}'_{1,3}\mathbf{x}'_0 + \mathbf{b}'_{1,3}) \\
& + \sigma(\mathbf{W}'_{1,4}\mathbf{x}'_0 + \mathbf{b}'_{1,4})] + \mathbf{b}'_{1,2}\} \\
& + \sigma\{\mathbf{W}'_{2,3}[\mathbf{W}'_{1,1}\mathbf{x}'_0 + \mathbf{b}'_{1,1} + \sigma(\mathbf{W}'_{1,2}\mathbf{x}'_0 + \mathbf{b}'_{1,2}) + \sigma(\mathbf{W}'_{1,3}\mathbf{x}'_0 + \mathbf{b}'_{1,3}) \\
& + \sigma(\mathbf{W}'_{1,4}\mathbf{x}'_0 + \mathbf{b}'_{1,4})] + \mathbf{b}'_{2,3}\} \\
& + \sigma\{\mathbf{W}'_{2,4}[\mathbf{W}'_{1,1}\mathbf{x}'_0 + \mathbf{b}'_{1,1} + \sigma(\mathbf{W}'_{1,2}\mathbf{x}'_0 + \mathbf{b}'_{1,2}) + \sigma(\mathbf{W}'_{1,3}\mathbf{x}'_0 + \mathbf{b}'_{1,3}) \\
& + \sigma(\mathbf{W}'_{1,4}\mathbf{x}'_0 + \mathbf{b}'_{1,4})] + \mathbf{b}'_{2,4}\} \\
= {} & (\mathbf{W}'_{2,1}\mathbf{W}'_{1,1}\mathbf{x}'_0 + \mathbf{W}'_{2,1}\mathbf{b}'_{1,1} + \mathbf{b}'_{2,1}) + \mathbf{W}'_{2,1}\sigma(\mathbf{W}'_{1,2}\mathbf{x}'_0 + \mathbf{b}'_{1,2}) + \mathbf{W}'_{2,1}\sigma(\mathbf{W}'_{1,3}\mathbf{x}'_0 + \mathbf{b}'_{1,3}) \\
& + \mathbf{W}'_{2,1}\sigma(\mathbf{W}'_{1,4}\mathbf{x}'_0 + \mathbf{b}'_{1,4}) \\
& + \sigma\{(\mathbf{W}'_{2,2}\mathbf{W}'_{1,1}\mathbf{x}'_0 + \mathbf{W}'_{2,2}\mathbf{b}'_{1,1} + \mathbf{b}'_{1,2}) + \mathbf{W}'_{2,2}\sigma(\mathbf{W}'_{1,2}\mathbf{x}'_0 + \mathbf{b}'_{1,2}) + \mathbf{W}'_{2,2}\sigma(\mathbf{W}'_{1,3}\mathbf{x}'_0 + \mathbf{b}'_{1,3}) \\
& + \mathbf{W}'_{2,2}\sigma(\mathbf{W}'_{1,4}\mathbf{x}'_0 + \mathbf{b}'_{1,4})\} \\
& + \sigma\{(\mathbf{W}'_{2,3}\mathbf{W}'_{1,1}\mathbf{x}'_0 + \mathbf{W}'_{2,3}\mathbf{b}'_{1,1} + \mathbf{b}'_{2,3}) + \mathbf{W}'_{2,3}\sigma(\mathbf{W}'_{1,2}\mathbf{x}'_0 + \mathbf{b}'_{1,2}) + \mathbf{W}'_{2,3}\sigma(\mathbf{W}'_{1,3}\mathbf{x}'_0 + \mathbf{b}'_{1,3}) \\
& + \mathbf{W}'_{2,3}\sigma(\mathbf{W}'_{1,4}\mathbf{x}'_0 + \mathbf{b}'_{1,4})\} \\
& + \sigma\{(\mathbf{W}'_{2,4}\mathbf{W}'_{1,1}\mathbf{x}'_0 + \mathbf{W}'_{2,4}\mathbf{b}'_{1,1} + \mathbf{b}'_{2,4}) + \mathbf{W}'_{2,4}\sigma(\mathbf{W}'_{1,2}\mathbf{x}'_0 + \mathbf{b}'_{1,2}) + \mathbf{W}'_{2,4}\sigma(\mathbf{W}'_{1,3}\mathbf{x}'_0 + \mathbf{b}'_{1,3}) \\
& + \mathbf{W}'_{2,4}\sigma(\mathbf{W}'_{1,4}\mathbf{x}'_0 + \mathbf{b}'_{1,4})\}
\end{aligned}
\tag{3}
$$

$$
\begin{aligned}
\mathbf{x}'_2 = {} & (\mathbf{W}'_{2,1}\mathbf{x}'_0 + \mathbf{b}'_{2,1}) + \mathbf{W}'_{2,1}\sigma(\mathbf{W}'_{1,2}\mathbf{x}'_0 + \mathbf{b}'_{1,2}) \\
& + \mathbf{W}'_{2,1}\sigma(\mathbf{W}'_{1,3}\mathbf{x}'_0 + \mathbf{b}'_{1,3}) + \mathbf{W}'_{2,1}\sigma(\mathbf{W}'_{1,4}\mathbf{x}'_0 + \mathbf{b}'_{1,4}) \\
& + \sigma(\mathbf{W}'_{2,2}\mathbf{x}'_0 + \mathbf{b}'_{2,2}) + \sigma(\mathbf{W}'_{2,3}\mathbf{x}'_0 + \mathbf{b}'_{2,3}) \\
& + \sigma(\mathbf{W}'_{2,4}\mathbf{x}'_0 + \mathbf{b}'_{2,4})
\end{aligned}
\tag{4}
$$

## C  THE DATA PROBLEM OF ILI

Figure 1 illustrates the data trends for the 7 features in the ILI dataset, labeled as Class 1 to Class 7. Time series data generally exhibit two main characteristics: periodicity and trends within each period. The seven features share a similar period; however, it is evident that Classes 6 and 7 show an upward trend over time, while the overall trends of the other features remain stable.

If there is any correlation between the features, we believe it would be either related to the period (which is the same across features and can thus be disregarded) or to the trend. Since Classes 6 and 7 both display a rising trend, while the trends of the other classes remain unchanged, we hypothesize that Classes 6 and 7 should be separated from the other features. Therefore, we grouped the data into two sets: one containing Classes 6 and 7, and the other with the remaining features. We then trained and tested the model on these separate groups. The test results, shown in Table 3, clearly indicate a significant improvement in model performance after grouping the data for training.

## D  THE PROPERTIES OF FOURIER TRANSFORM

Compared to convolution, learning features in the Fourier domain offers the advantage of a larger receptive field. This phenomenon is illustrated in Figures 2 and 3. Figure 2 shows a typical convolu-

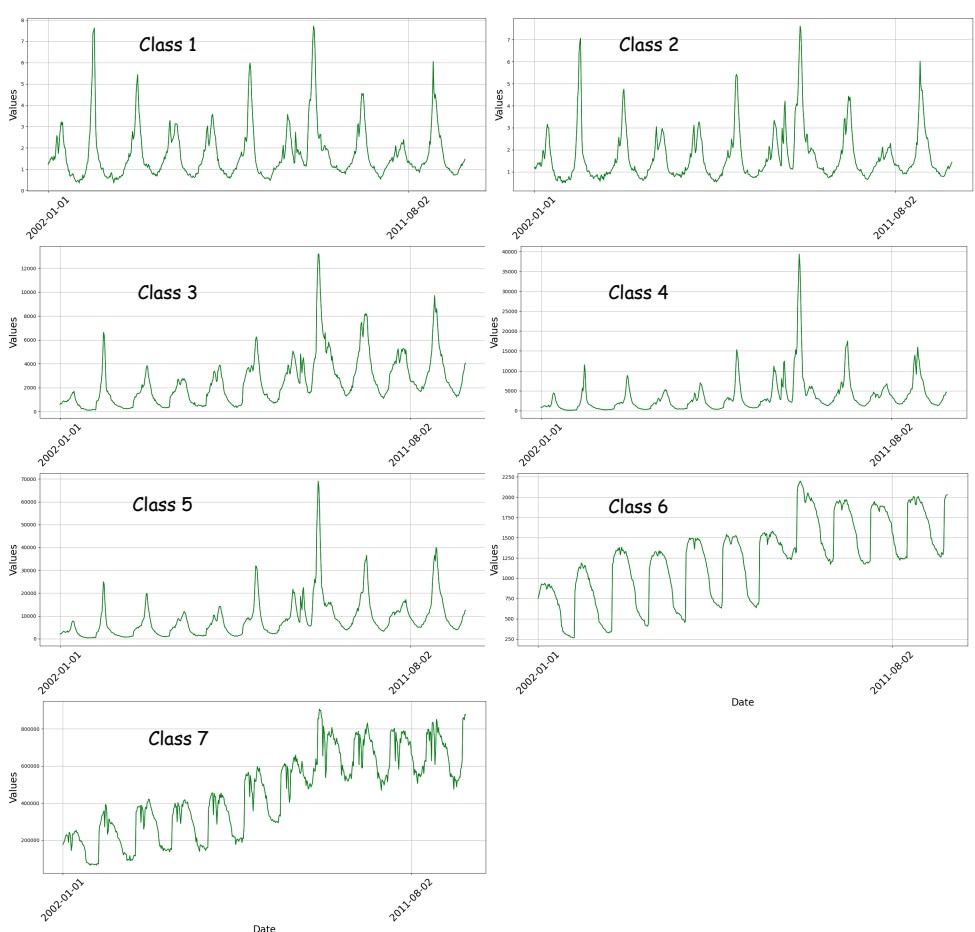

Figure 1: The trends of the features in the ILI dataset.

Table 3: Based on the characteristics of the ILI data, we divided it into two groups: Group 1: Class 1, 2, 3, 4, 5; Group 2: Class 6, 7. We then calculated the MSE and MAE for each group separately. Ori: Results from training with all features together. Split: Results from training after splitting the features into two groups.

| Methods | Metric | Class 1 MSE | Class 1 MAE | Class 2 MSE | Class 2 MAE | Class 3 MSE | Class 3 MAE | Class 4 MSE | Class 4 MAE | Class 5 MSE | Class 5 MAE | Class 6 MSE | Class 6 MAE | Class 7 MSE | Class 7 MAE |
|---|---|---|---|---|---|---|---|---|---|---|---|---|---|---|---|
| Ori | 24 | 0.532 | 0.764 | 0.602 | 0.987 | 1.085 | 2.527 | 0.968 | 2.501 | 1.196 | 3.635 | 0.736 | 0.821 | 0.876 | 1.062 |
| | 36 | 0.630 | 0.981 | 0.656 | 1.149 | 1.149 | 2.908 | 1.052 | 3.075 | 1.364 | 4.744 | 0.689 | 0.695 | 0.868 | 1.042 |
| | 48 | 0.596 | 0.891 | 0.644 | 1.063 | 1.129 | 2.801 | 1.035 | 2.867 | 1.266 | 4.356 | 0.727 | 0.763 | 0.892 | 1.063 |
| | 60 | 0.566 | 0.807 | 0.621 | 0.986 | 1.193 | 3.051 | 1.025 | 2.785 | 1.250 | 4.210 | 0.824 | 0.922 | 0.980 | 1.222 |
| | Avg | **0.581** | **0.860** | **0.630** | **1.046** | **1.139** | **2.821** | **1.020** | **2.807** | **1.269** | **4.236** | **0.744** | **0.800** | **0.904** | **1.097** |
| Split | 24 | 0.402 | 0.284 | 0.372 | 0.260 | 0.308 | 0.178 | 0.360 | 0.292 | 0.412 | 0.334 | 0.154 | 0.204 | 0.047 | 0.079 |
| | 36 | 0.350 | 0.217 | 0.326 | 0.206 | 0.289 | 0.160 | 0.275 | 0.146 | 0.343 | 0.210 | 0.182 | 0.060 | 0.221 | 0.091 |
| | 48 | 0.370 | 0.229 | 0.339 | 0.211 | 0.299 | 0.157 | 0.309 | 0.199 | 0.368 | 0.251 | 0.207 | 0.072 | 0.240 | 0.103 |
| | 60 | 0.342 | 0.205 | 0.320 | 0.206 | 0.281 | 0.147 | 0.275 | 0.159 | 0.345 | 0.222 | 0.211 | 0.076 | 0.231 | 0.103 |
| | Avg | **0.366** | **0.233** | **0.339** | **0.220** | **0.294** | **0.160** | **0.304** | **0.199** | **0.367** | **0.254** | **0.1885** | **0.103** | **0.184** | **0.094** |

tion process with a kernel size of 3, resulting in a receptive field of 3. In contrast, the black boxes of Figure 3 depict a standard Fourier transform process, while the right side (highlighted in the red box) demonstrates learning in the Fourier space based on the Fourier transform results from the left. It is evident that every point within the red box on the right is capable of gathering global information from the original domain, thus offering an expanded receptive field compared to convolution.

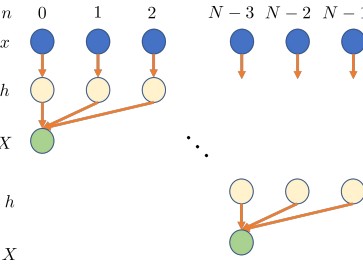

Figure 2: The convolution operation on the sequence. $n$ is the serial number. $\mathbf{x}$ is a sequence. $\mathbf{h}$ is the convolutional kernel and the kernel size is three. $\mathbf{X}$ is the output after convulution.

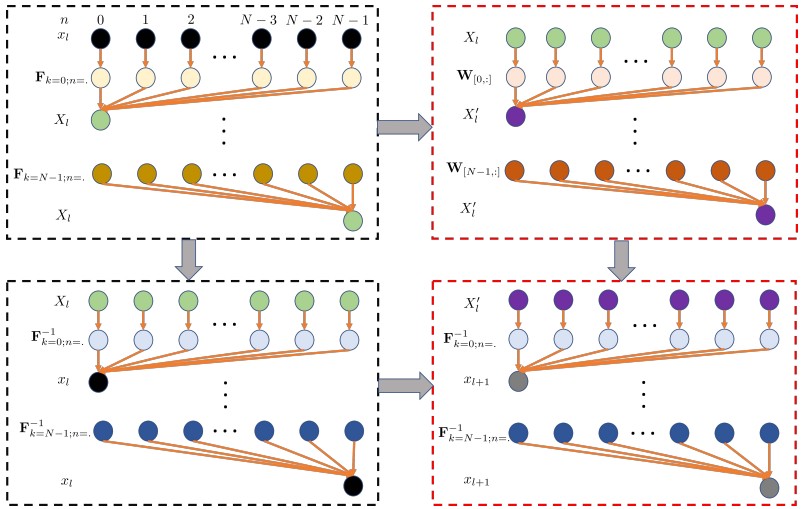

Figure 3: The left part is the Fourier transform. $n$ is the serial number and $n = .$ represents $n = 0, 1, 2, ....$ $\mathbf{x}_l$ and $\mathbf{x}_{l+1}$ are latent sequences. $\mathbf{F}$ is the FT matrix and $\mathbf{F}^{-1}$ is inverse matrix of the FT. The right part is the way of learning the change of sequence in the Fourier domain. $\mathbf{W}$ is the weight matrix. $\mathbf{X}_l$ is the FT output. $\mathbf{X}'_l$ is the latent features learned in the Fourier domain.

# E  VISUALIZATION OF WEIGHTS

Figures 4, 5, 6, and 7 illustrate the visual comparison of parameter weight information within the second MRF block of the MRFNet model on ETTh1 dataset, along with the weight information at corresponding positions for LS, LC, and LF. Overall, from the texture of these images, it can be observed that they learn similar information.

**Linear Block:** The overall texture learned by the Linear Block is similar, yet the weight parameter magnitudes differ among the models. Regarding the matrix multiplication, the x-axis corresponds to input time information, while the y-axis corresponds to output time information. In the row direction, there is an overall appearance of an interlocking grid pattern. This pattern signifies that different positions of input have varying degrees of importance for the output, and this importance changes continuously. Among these models, the LS model's parameters exhibit this characteristic most prominently. This is primarily due to the LS model learning a sparse matrix, implying a block-wise learning of temporal information. Therefore, the linear Block complements this block-wise feature, reinforcing the coordination of local and global information. The other models learn information holistically, which is why this characteristic is less evident.

**Sparse Matrix Block:** The Sparse Matrix Block demonstrates a similar texture of weight features as the linear Block, with the distinction that this pattern is block-wise. The MRFNet model learns more discrete feature information, whereas the LS model learns continuous, block-wise information. However, the underlying patterns they learn are similar.

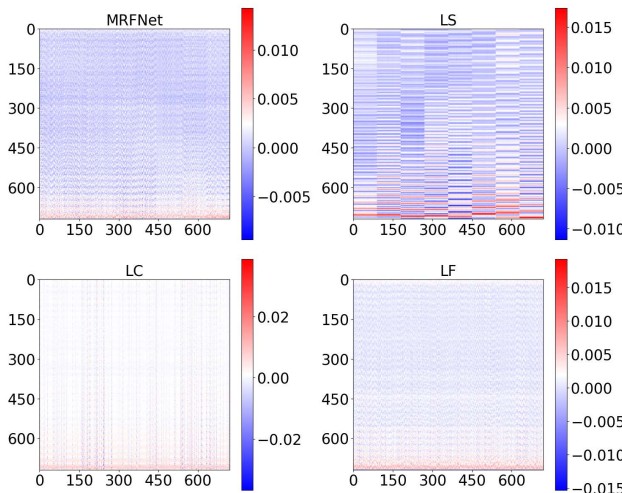

Figure 4: The images depict the weights of the Linear Block for the same layer corresponding to MRFNet, LS, LC, and LF.

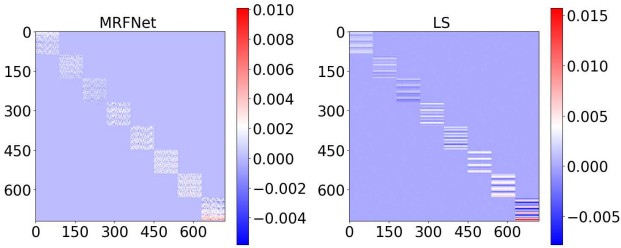

Figure 5: The images depict the weights of the Sparse Matrix Block for the same layer corresponding to MRFNet and LS.

**Convolutional Block:** Given a kernel size of 3, the display of convolutional kernel weights is organized according to the kernel positions. It is evident that the textures they learn are highly alike.

**FT Block:** The FT Block's weights consist of real and imaginary components, each displayed separately. The similarity of the textures is conspicuous.

In summary, the weights corresponding to LS, LC, and LF in the MRFNet model are notably similar. This observation indicates that they collectively learn akin information, which is coherent due to being trained on the same dataset. Additionally, as the MRFNet model comprises multiple modules, these modules function complementarily. Consequently, specific differences emerge among these modules.

## F    PROFERMANCES OF UNIVARIATE PREDICTION

We compared preferences for univariate prediction in Table 4. Overall, our results are similar to the PatchTST and outperform other models.

## G    THE PREDICTION RESULTS OF MRFNET

Figure 8 presents the prediction results of the HUFL feature in the ETTm1 dataset. As shown in the figure, our model outperforms the PatchTST, DLinear, and Autoformer models.

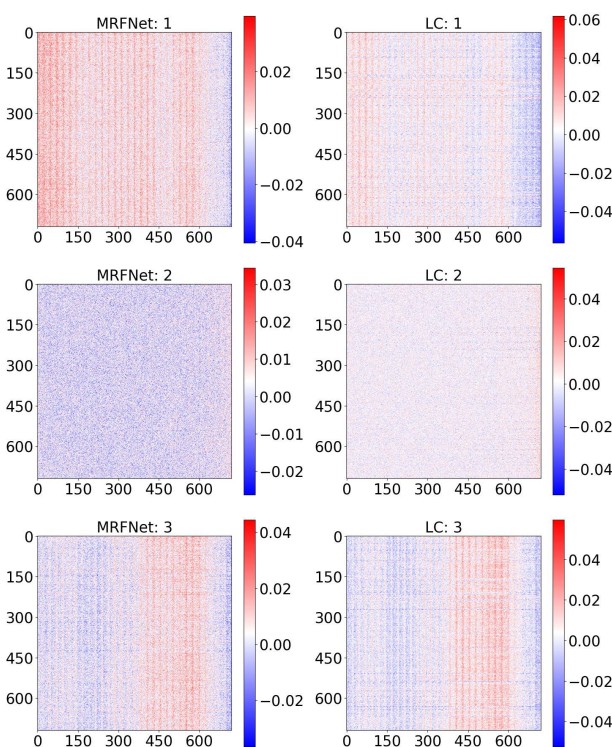

Figure 6: The images depict the weights of the Conv Block for the same layer corresponding to MRFNet and LC. The input and output dimensions are 720 and the kernel size is three, so the dimensions of the kennel are $(720, 720, 3)$. In this figure, 1, 2, and 3 denote the number of the third dimension direction of $(720, 720, 3)$.

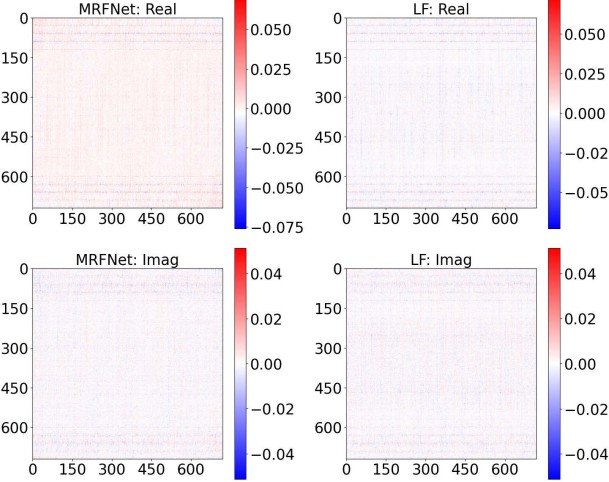

Figure 7: The images depict the weights of the Fourier Block for the same layer corresponding to MRFNet and LF. Real and Imag represent the real and imaginary parts of the weights in Fourier domain.

Table 4: Univariate predictions of ETTh1, ETTh2, ETTm1, and ETTm2 by twelve models.

| Models | | MRFNet | | PatchTST/64 | | DLinear | | FEDformer | | Autoformer | | Informer | | LogTrans | |
|---|---|---|---|---|---|---|---|---|---|---|---|---|---|---|---|
| Metric | | MSE | MAE | MSE | MAE | MSE | MAE | MSE | MAE | MSE | MAE | MSE | MAE | MSE | MAE |
| ETTh1 | 96 | **0.055** | **0.181** | 0.059 | 0.189 | 0.056 | 0.180 | 0.079 | 0.215 | 0.071 | 0.206 | 0.193 | 0.377 | 0.283 | 0.468 |
| | 192 | **0.071** | **0.206** | 0.074 | 0.215 | **0.071** | **0.204** | 0.104 | 0.245 | 0.114 | 0.262 | 0.217 | 0.395 | 0.234 | 0.409 |
| | 336 | **0.084** | **0.231** | **0.076** | **0.220** | 0.098 | 0.244 | 0.119 | 0.270 | 0.107 | 0.258 | 0.202 | 0.381 | 0.386 | 0.546 |
| | 720 | **0.098** | **0.246** | **0.087** | **0.236** | 0.189 | 0.359 | 0.142 | 0.299 | 0.126 | 0.283 | 0.183 | 0.355 | 0.475 | 0.629 |
| ETTh2 | 96 | **0.151** | 0.301 | **0.131** | **0.284** | **0.131** | **0.279** | 0.128 | 0.271 | 0.153 | 0.306 | 0.213 | 0.373 | 0.217 | 0.379 |
| | 192 | **0.171** | **0.329** | **0.171** | **0.329** | **0.176** | **0.329** | 0.185 | **0.330** | 0.204 | 0.351 | 0.227 | 0.387 | 0.281 | 0.429 |
| | 336 | **0.186** | 0.345 | **0.171** | **0.336** | 0.209 | 0.367 | 0.231 | 0.378 | 0.246 | 0.389 | 0.242 | 0.401 | 0.293 | 0.437 |
| | 720 | **0.214** | **0.368** | **0.223** | **0.380** | 0.276 | 0.426 | 0.278 | 0.420 | 0.268 | 0.409 | 0.291 | 0.439 | 0.218 | 0.387 |
| ETTm1 | 96 | **0.026** | **0.121** | **0.026** | **0.123** | 0.028 | 0.123 | 0.033 | 0.140 | 0.056 | 0.183 | 0.109 | 0.277 | 0.049 | 0.171 |
| | 192 | **0.041** | **0.153** | **0.040** | **0.151** | 0.045 | 0.156 | 0.058 | 0.186 | 0.081 | 0.216 | 0.151 | 0.310 | 0.157 | 0.317 |
| | 336 | **0.055** | **0.178** | **0.053** | **0.174** | 0.061 | 0.182 | 0.084 | 0.231 | 0.076 | 0.218 | 0.427 | 0.591 | 0.289 | 0.459 |
| | 720 | **0.070** | **0.203** | 0.073 | 0.206 | 0.080 | 0.210 | 0.102 | 0.250 | 0.110 | 0.267 | 0.438 | 0.586 | 0.430 | 0.579 |
| ETTm2 | 96 | **0.063** | **0.181** | 0.065 | 0.187 | **0.063** | **0.183** | 0.067 | 0.198 | 0.065 | 0.189 | 0.088 | 0.225 | 0.075 | 0.208 |
| | 192 | **0.091** | **0.226** | 0.093 | 0.231 | **0.092** | **0.227** | 0.102 | 0.245 | 0.118 | 0.256 | 0.132 | 0.283 | 0.129 | 0.275 |
| | 336 | 0.128 | 0.272 | **0.121** | **0.266** | **0.119** | **0.261** | 0.130 | 0.279 | 0.154 | 0.305 | 0.180 | 0.336 | 0.154 | 0.302 |
| | 720 | **0.165** | **0.315** | **0.172** | 0.322 | 0.175 | **0.320** | 0.178 | 0.325 | 0.182 | 0.335 | 0.300 | 0.435 | 0.160 | 0.321 |

Table 5: Multivariate predictions of ETTh1, ETTh2, ETTm1, ETTm2, Traffic, Electricity, Weather and ILI, by twelve models. The best results are highlighted in bold red. The second-best results are indicated with highlighted in bold black.

| Methods | | MRFNet | | GPT2(6) | | DLinear | | PatchTST | | TimesNet | | FEDformer | | Autoformer | | Stationary | | ETSformer | | LightTS | | Informer | | Reformer | |
|---|---|---|---|---|---|---|---|---|---|---|---|---|---|---|---|---|---|---|---|---|---|---|---|---|---|
| Metric | | MSE | MAE | MSE | MAE | MSE | MAE | MSE | MAE | MSE | MAE | MSE | MAE | MSE | MAE | MSE | MAE | MSE | MAE | MSE | MAE | MSE | MAE | MSE | MAE |
| Weather | 96 | 0.149 | 0.191 | 0.162 | 0.212 | 0.176 | 0.237 | 0.149 | 0.198 | 0.172 | 0.220 | 0.217 | 0.296 | 0.266 | 0.336 | 0.173 | 0.223 | 0.197 | 0.281 | 0.182 | 0.242 | 0.300 | 0.384 | 0.689 | 0.596 |
| | 192 | 0.193 | 0.235 | 0.204 | 0.248 | 0.220 | 0.282 | 0.194 | 0.241 | 0.219 | 0.261 | 0.276 | 0.336 | 0.307 | 0.367 | 0.245 | 0.285 | 0.237 | 0.312 | 0.227 | 0.287 | 0.598 | 0.544 | 0.752 | 0.638 |
| | 336 | 0.245 | 0.278 | 0.254 | 0.286 | 0.265 | 0.319 | 0.245 | 0.282 | 0.280 | 0.306 | 0.339 | 0.380 | 0.359 | 0.395 | 0.321 | 0.338 | 0.298 | 0.353 | 0.282 | 0.334 | 0.578 | 0.523 | 0.639 | 0.596 |
| | 720 | 0.315 | 0.329 | 0.326 | 0.337 | 0.333 | 0.362 | 0.314 | 0.334 | 0.365 | 0.359 | 0.403 | 0.428 | 0.419 | 0.428 | 0.414 | 0.410 | 0.352 | 0.288 | 0.352 | 0.386 | 1.059 | 0.741 | 1.130 | 0.792 |
| | Avg | **0.225** | **0.258** | 0.237 | 0.270 | 0.248 | 0.300 | **0.225** | 0.264 | 0.259 | 0.287 | 0.309 | 0.360 | 0.338 | 0.382 | 0.288 | 0.314 | 0.271 | 0.334 | 0.261 | 0.312 | 0.634 | 0.548 | 0.803 | 0.656 |
| ETTh1 | 96 | 0.364 | 0.393 | 0.376 | 0.397 | 0.375 | 0.399 | 0.370 | 0.399 | 0.384 | 0.402 | 0.376 | 0.419 | 0.449 | 0.459 | 0.513 | 0.491 | 0.494 | 0.479 | 0.424 | 0.432 | 0.865 | 0.713 | 0.837 | 0.728 |
| | 192 | 0.402 | 0.415 | 0.416 | 0.418 | 0.405 | 0.416 | 0.413 | 0.421 | 0.436 | 0.429 | 0.420 | 0.448 | 0.500 | 0.482 | 0.534 | 0.504 | 0.538 | 0.504 | 0.475 | 0.462 | 1.008 | 0.792 | 0.923 | 0.766 |
| | 336 | 0.442 | 0.444 | 0.442 | 0.433 | 0.439 | 0.443 | 0.422 | 0.436 | 0.491 | 0.469 | 0.459 | 0.465 | 0.521 | 0.496 | 0.588 | 0.535 | 0.574 | 0.521 | 0.518 | 0.488 | 1.107 | 0.809 | 1.097 | 0.835 |
| | 720 | 0.434 | 0.454 | 0.477 | 0.456 | 0.472 | 0.490 | 0.447 | 0.466 | 0.521 | 0.500 | 0.506 | 0.507 | 0.514 | 0.512 | 0.643 | 0.616 | 0.562 | 0.535 | 0.547 | 0.533 | 1.181 | 0.865 | 1.257 | 0.889 |
| | Avg | **0.410** | **0.426** | 0.427 | 0.426 | 0.422 | 0.437 | **0.413** | 0.430 | 0.458 | 0.450 | 0.440 | 0.460 | 0.496 | 0.487 | 0.570 | 0.537 | 0.542 | 0.510 | 0.491 | 0.479 | 1.040 | 0.795 | 1.029 | 0.805 |
| ETTh2 | 96 | 0.273 | 0.330 | 0.285 | 0.342 | 0.289 | 0.353 | 0.274 | 0.336 | 0.340 | 0.374 | 0.358 | 0.397 | 0.346 | 0.388 | 0.476 | 0.458 | 0.340 | 0.391 | 0.397 | 0.437 | 3.755 | 1.525 | 2.626 | 1.317 |
| | 192 | 0.341 | 0.376 | 0.354 | 0.389 | 0.383 | 0.418 | 0.339 | 0.379 | 0.402 | 0.414 | 0.429 | 0.439 | 0.456 | 0.452 | 0.512 | 0.493 | 0.430 | 0.439 | 0.520 | 0.504 | 5.602 | 1.931 | 11.12 | 2.979 |
| | 336 | 0.366 | 0.396 | 0.373 | 0.407 | 0.448 | 0.465 | 0.329 | 0.380 | 0.452 | 0.452 | 0.496 | 0.487 | 0.482 | 0.486 | 0.552 | 0.551 | 0.485 | 0.479 | 0.626 | 0.559 | 9.323 | 1.835 | 9.323 | 2.769 |
| | 720 | 0.385 | 0.423 | 0.406 | 0.441 | 0.605 | 0.551 | 0.379 | 0.422 | 0.462 | 0.468 | 0.463 | 0.474 | 0.515 | 0.511 | 0.562 | 0.560 | 0.500 | 0.497 | 0.863 | 0.672 | 3.647 | 1.625 | 3.874 | 1.697 |
| | Avg | **0.341** | **0.381** | 0.354 | 0.394 | 0.431 | 0.446 | **0.330** | **0.379** | 0.414 | 0.427 | 0.437 | 0.449 | 0.450 | 0.459 | 0.526 | 0.516 | 0.439 | 0.452 | 0.602 | 0.543 | 4.431 | 1.729 | 6.736 | 2.191 |
| ETTm1 | 96 | 0.297 | 0.342 | 0.292 | 0.346 | 0.299 | 0.343 | 0.290 | 0.342 | 0.338 | 0.375 | 0.379 | 0.419 | 0.505 | 0.475 | 0.386 | 0.398 | 0.375 | 0.398 | 0.374 | 0.400 | 0.672 | 0.571 | 0.538 | 0.528 |
| | 192 | 0.334 | 0.366 | 0.332 | 0.372 | 0.335 | 0.365 | 0.332 | 0.369 | 0.374 | 0.387 | 0.426 | 0.441 | 0.553 | 0.496 | 0.459 | 0.444 | 0.408 | 0.410 | 0.400 | 0.407 | 0.795 | 0.669 | 0.658 | 0.592 |
| | 336 | 0.366 | 0.385 | 0.366 | 0.394 | 0.369 | 0.386 | 0.366 | 0.392 | 0.410 | 0.411 | 0.445 | 0.459 | 0.621 | 0.537 | 0.495 | 0.464 | 0.435 | 0.428 | 0.438 | 0.438 | 1.212 | 0.871 | 0.898 | 0.721 |
| | 720 | 0.407 | 0.411 | 0.417 | 0.421 | 0.425 | 0.421 | 0.416 | 0.420 | 0.478 | 0.450 | 0.543 | 0.490 | 0.671 | 0.561 | 0.585 | 0.516 | 0.499 | 0.462 | 0.527 | 0.502 | 1.166 | 0.823 | 1.102 | 0.841 |
| | Avg | **0.351** | **0.376** | 0.352 | 0.383 | 0.357 | 0.378 | **0.351** | 0.380 | 0.400 | 0.406 | 0.448 | 0.452 | 0.588 | 0.517 | 0.481 | 0.456 | 0.429 | 0.425 | 0.435 | 0.437 | 0.961 | 0.734 | 0.799 | 0.671 |
| ETTm2 | 96 | 0.163 | 0.246 | 0.173 | 0.262 | 0.167 | 0.269 | 0.165 | 0.255 | 0.187 | 0.267 | 0.203 | 0.287 | 0.255 | 0.339 | 0.192 | 0.274 | 0.189 | 0.280 | 0.209 | 0.308 | 0.365 | 0.453 | 0.658 | 0.619 |
| | 192 | 0.219 | 0.287 | 0.229 | 0.301 | 0.224 | 0.303 | 0.220 | 0.292 | 0.249 | 0.309 | 0.269 | 0.328 | 0.281 | 0.340 | 0.280 | 0.339 | 0.253 | 0.319 | 0.311 | 0.382 | 0.533 | 0.563 | 1.078 | 0.827 |
| | 336 | 0.275 | 0.323 | 0.286 | 0.341 | 0.281 | 0.342 | 0.274 | 0.329 | 0.321 | 0.351 | 0.325 | 0.366 | 0.339 | 0.372 | 0.334 | 0.361 | 0.314 | 0.357 | 0.442 | 0.466 | 1.363 | 0.887 | 1.549 | 0.972 |
| | 720 | 0.354 | 0.377 | 0.378 | 0.401 | 0.397 | 0.421 | 0.362 | 0.385 | 0.408 | 0.403 | 0.421 | 0.415 | 0.433 | 0.432 | 0.417 | 0.413 | 0.414 | 0.413 | 0.675 | 0.587 | 3.379 | 1.338 | 2.631 | 1.242 |
| | Avg | **0.252** | **0.333** | 0.266 | 0.326 | 0.267 | 0.333 | **0.255** | 0.315 | 0.291 | 0.333 | 0.305 | 0.349 | 0.327 | 0.371 | 0.306 | 0.347 | 0.293 | 0.342 | 0.409 | 0.436 | 1.410 | 0.810 | 1.479 | 0.915 |
| ILI | 24 | 1.757 | 0.857 | 2.063 | 0.881 | 2.215 | 1.081 | 1.319 | 0.754 | 2.317 | 0.934 | 3.228 | 1.260 | 3.483 | 1.287 | 2.294 | 0.945 | 2.527 | 1.020 | 8.313 | 2.144 | 5.764 | 1.677 | 4.400 | 1.382 |
| | 36 | 2.085 | 0.915 | 1.868 | 0.892 | 1.963 | 0.963 | 1.430 | 0.834 | 1.972 | 0.920 | 2.679 | 1.080 | 3.103 | 1.148 | 1.825 | 0.848 | 2.615 | 1.007 | 6.631 | 1.902 | 4.755 | 1.467 | 4.783 | 1.448 |
| | 48 | 1.972 | 0.899 | 1.790 | 0.884 | 2.130 | 1.024 | 1.553 | 0.815 | 2.238 | 0.940 | 2.622 | 1.078 | 2.669 | 1.085 | 2.010 | 0.900 | 2.359 | 0.972 | 7.299 | 1.982 | 4.763 | 1.469 | 4.832 | 1.465 |
| | 60 | 1.998 | 0.923 | 1.979 | 0.957 | 2.368 | 1.096 | 1.470 | 0.788 | 2.027 | 0.928 | 2.857 | 1.157 | 2.770 | 1.125 | 2.178 | 0.963 | 2.487 | 1.016 | 7.283 | 1.985 | 5.264 | 1.564 | 4.882 | 1.483 |
| | Avg | 1.953 | **0.898** | **1.925** | 0.903 | 2.169 | 1.041 | **1.443** | **0.797** | 2.139 | 0.931 | 2.847 | 1.144 | 3.006 | 1.161 | 2.077 | 0.914 | 2.497 | 1.004 | 7.382 | 2.003 | 5.137 | 1.544 | 4.724 | 1.445 |
| ECL | 96 | 0.127 | 0.218 | 0.139 | 0.238 | 0.140 | 0.237 | 0.129 | 0.222 | 0.168 | 0.272 | 0.193 | 0.308 | 0.201 | 0.317 | 0.169 | 0.273 | 0.187 | 0.304 | 0.207 | 0.307 | 0.274 | 0.368 | 0.312 | 0.402 |
| | 192 | 0.144 | 0.234 | 0.153 | 0.251 | 0.153 | 0.249 | 0.157 | 0.240 | 0.184 | 0.289 | 0.201 | 0.315 | 0.222 | 0.334 | 0.182 | 0.286 | 0.199 | 0.315 | 0.213 | 0.316 | 0.296 | 0.386 | 0.348 | 0.433 |
| | 336 | 0.159 | 0.251 | 0.169 | 0.266 | 0.169 | 0.267 | 0.163 | 0.259 | 0.198 | 0.300 | 0.214 | 0.329 | 0.231 | 0.338 | 0.200 | 0.304 | 0.212 | 0.329 | 0.230 | 0.333 | 0.300 | 0.394 | 0.350 | 0.433 |
| | 720 | 0.192 | 0.280 | 0.206 | 0.297 | 0.203 | 0.301 | 0.197 | 0.290 | 0.220 | 0.320 | 0.246 | 0.355 | 0.254 | 0.361 | 0.222 | 0.321 | 0.233 | 0.345 | 0.265 | 0.360 | 0.373 | 0.439 | 0.340 | 0.420 |
| | Avg | **0.155** | **0.245** | 0.167 | 0.263 | 0.166 | 0.263 | **0.161** | 0.252 | 0.192 | 0.295 | 0.214 | 0.327 | 0.227 | 0.338 | 0.193 | 0.296 | 0.208 | 0.323 | 0.229 | 0.329 | 0.311 | 0.397 | 0.338 | 0.422 |
| Traffic | 96 | 0.386 | 0.241 | 0.388 | 0.282 | 0.410 | 0.282 | 0.360 | 0.249 | 0.593 | 0.321 | 0.587 | 0.366 | 0.613 | 0.388 | 0.612 | 0.338 | 0.607 | 0.392 | 0.615 | 0.391 | 0.719 | 0.391 | 0.732 | 0.423 |
| | 192 | 0.399 | 0.248 | 0.407 | 0.290 | 0.423 | 0.287 | 0.379 | 0.256 | 0.617 | 0.336 | 0.604 | 0.373 | 0.616 | 0.382 | 0.613 | 0.340 | 0.621 | 0.399 | 0.601 | 0.382 | 0.696 | 0.379 | 0.733 | 0.420 |
| | 336 | 0.414 | 0.274 | 0.412 | 0.294 | 0.436 | 0.296 | 0.392 | 0.264 | 0.629 | 0.336 | 0.621 | 0.383 | 0.622 | 0.337 | 0.618 | 0.328 | 0.622 | 0.396 | 0.613 | 0.386 | 0.777 | 0.420 | 0.742 | 0.420 |
| | 720 | 0.457 | 0.277 | 0.450 | 0.312 | 0.466 | 0.315 | 0.432 | 0.286 | 0.640 | 0.350 | 0.626 | 0.382 | 0.660 | 0.408 | 0.653 | 0.355 | 0.632 | 0.396 | 0.658 | 0.407 | 0.864 | 0.472 | 0.755 | 0.423 |
| | Avg | **0.414** | **0.260** | 0.414 | 0.294 | 0.433 | 0.295 | **0.390** | 0.263 | 0.620 | 0.336 | 0.610 | 0.376 | 0.628 | 0.379 | 0.624 | 0.340 | 0.621 | 0.396 | 0.622 | 0.392 | 0.764 | 0.416 | 0.741 | 0.422 |

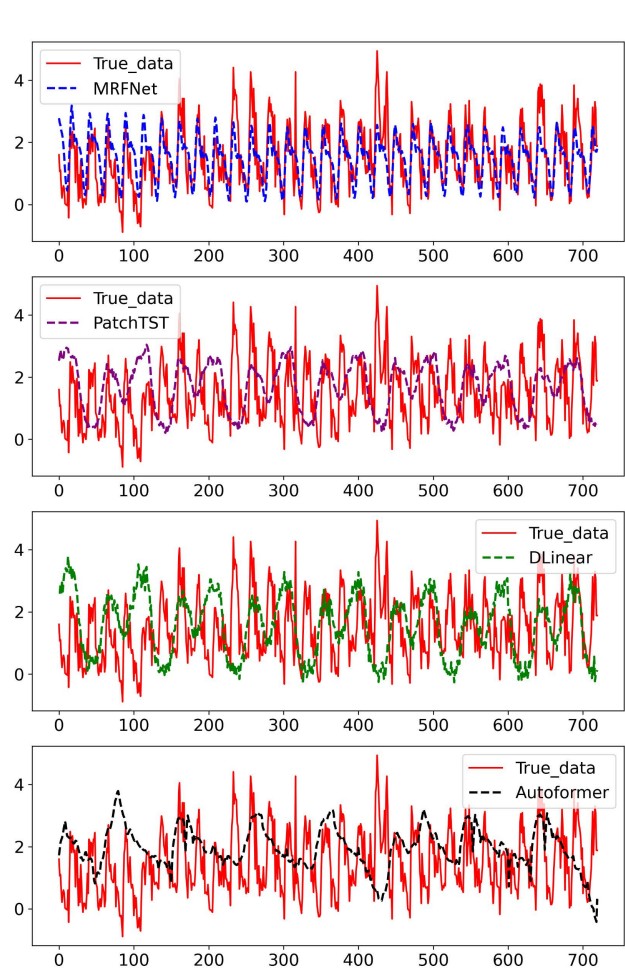

Figure 8: The prediction results (Horizon = 720; HUFL) of MRFNet, PatchTST, DLinear, Auto-former on the ETTm1 dataset.

