pling the periodicity and trend in time series data. Dai et al. (2024) propose a novel Periodicity Decoupling Framework (PDF) to capture 2D temporal variations of decoupled series for long-term series forecasting.

**Transformer**: Transformers have dominated deep learning and shown significant potential in solving time series forecasting problems (Li et al., 2019; Wu et al., 2020; Lim et al., 2021; Wen et al., 2022). The multi-head attention architecture can extract information while positional embeddings help retain sequence position information (Kitaev et al., 2020; Zhang and Zhu, 2018; Wu et al., 2021; Shen and Wang, 2022; Madhusudhanan et al., 2021). However, Transformers are computationally complex, and the setting of hyperparameters greatly influences the performance of Transformer-based models. To address these issues, models like Informer, Autoformer, and Fedformer were developed (Zhou et al., 2021; Wu et al., 2021; Zhou et al., 2022). Liu et al. (2022) tackles time series problems from the perspective of stationarity, while ETSformer (Woo et al., 2022) uses exponential smoothing and frequency attention to replace the self-attention mechanism in Transformers, enhancing accuracy and efficiency. Wang et al. (2023) proposed a Channel Aligned Robust Blend Transformer to address the shortcomings of channel correlation.

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

## 5.2 DATASETS

The experiments were conducted on real-world datasets (Zhou et al., 2021), including (1) Electricity Transformer Temperature (ETT), (2) Electricity, and (3) Traffic, (4) Weather, (5) ILI. The details of all datasets can be found in (Wu et al., 2021). The data source is available at github[1]. It should be noted that ETT consists of four different datasets: ETTh1, ETTh2, ETTm1, and ETTm2, each of which has seven variables. We evaluate our model using Mean Absolute Errors (MAE) and Mean

---
[1]https://github.com/thuml/Autoformer

Squared Errors (MSE), as used in (Zhou et al., 2021). Smaller values of MAE/MSE indicate better model performance. We use the average values for all predictions. The details of all datasets are shown in Table 1. In Appendix A, we give the data setting for training, evaluation and testing.

## 5.3 Results of MRFNet

In Table 2, we compare our model with the current SOTA models for time series prediction. Our model demonstrates superior performances on these datasets, achieving SOTA results in most cases (except for the ILI dataset, which is primarily affected by the characteristics of the data itself; we will provide a solution for this issue in Section 5.4). However, when analyzing the results across all models, it becomes evident that the current SOTA models - MRFNet, GPT2(6) (Zhou et al., 2023), DLinear (Zeng et al., 2023), and PatchTST (Nie et al., 2022) - converge towards a similar performance bottleneck. This is because they are all composed of multi-layer Transformers and Linear components, which we have proven to be specific implementations of the dynamic UAT, leading them to the same blockneck with limited data. Additional results for univariate prediction can be found in Appendix F.

Although the MRFNet proposed in this paper is specifically designed to leverage the characteristics of temporal data, it remains categorized within the UAT function framework. Consequently, MRFNet provides only limited improvements over previous prediction results. However, experimental outcomes demonstrate that the model achieves SOTA performance, thereby validating its effectiveness. We believe that another critical factor contributing to the existing performance bottleneck lies in the inherent limitations of the data itself. To address this, in Section 5.4, we conducted more in-depth data processing and training optimizations tailored to the unique properties of temporal data.

## 5.4 Data Effects

In this section, we will solve the bottleneck problem from the perspective of data. A crucial characteristic of time series data is its periodicity. However, this periodicity may change over time due to external factors (e.g., advancements in science leading to a gradual increase in electricity demand). This gives data a specific property: data points closer to the prediction point have a greater impact on the results (e.g., when forecasting electricity demand for 2024, the data from 2023 is more relevant, while earlier years are less so).

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

These results support our conclusions: first, we have demonstrated that these models inherently share the same mathematical properties, which explains the similarity in their results. Second, MRFNet is equivalent to an increased network depth compared to the other models and benefits from its more comprehensive feature learning. As a result, MRFNet maintains SOTA performance or near-SOTA performance across all datasets, further validating its robustness and superiority.

Additionally, in Appendix E, we compare the weights of MRFNet, LS, LC, and LF at the same layer. The weights for the same dataset exhibit similar textures, suggesting that, in general, they are learning similar information patterns. Given that we have proven these models share the same mathematical framework, the primary differences lie in the size of their receptive fields, with each model's module playing a similar role. However, due to the different receptive field sizes, the specific patterns they learn may vary slightly, though the overall trend remains consistent.

## 6 CONCLUSION

In this paper, we demonstrate that most models in the field of time series forecasting are specific implementations of the UAT, which explains why current SOTA models in time series forecasting tend to converge towards the same performance bottleneck. Based on the principles of UAT and the characteristics of time series data, we have designed a new model, MRFNet, for time series prediction. MRFNet integrates linear modules, sparse matrix modules, convolutional modules, and Fourier transform modules, effectively capturing both global and local receptive field information. Through extensive testing on various common datasets, the MRFNet model has demonstrated its superiority, achieving SOTA-level performance. Additionally, by leveraging the intrinsic properties of time series data, we further refined the data, significantly enhancing the performance on certain datasets. Finally, we conducted experiments to confirm that time series forecasting models based on UAT theory eventually converge to a similar performance bottleneck. In the future, we will further try to tackle the challenges of bottleneck from the perspective of UAT and data.