# OpenReview forum: "MRFNet: Multi-Receptive Field  Network for Multivariate Time-Series Prediction"
_ICLR.cc/2025/Conference — ICLR 2025 Conference Withdrawn Submission_

### Official Review · Reviewer_T3gY · 2024-10-21

**Soundness:** 3
**Presentation:** 2
**Contribution:** 2
**Rating:** 5
**Confidence:** 3

**Summary:**

This paper is inspired by the universal approximation theorem (UAT) to solve the time-series forecasting problem. It shows that most time-series forecasting models can be expressed in the matrix-vector form thus satisfying the UAT. It also proposes the multi-receptive field network (MRFNet) for time-series forecasting problems. MRFNet consists of four blocks linear, sparse matrix, convolutional and Fourier transform blocks claimed to be interpretable and to capture both global and local information.

**Strengths:**

1) The problem of time-series forecasting is important because it has many real-world applications.
2) the concept of universal approximation theorem is theoretically sound.

**Weaknesses:**

Motivation

motivation of this work is rather unclear. It does not point out a specific issue to be addressed in the paper. Rather, the use of UAT is a common issue. In addition, please clarify the fact that neural network is a universal approximator. On the other hand, there is no figures at all justifying the motivation of this paper.

Literature Survey

literature review is rather outdated. There are more recent works uncovered in the literature survey.

Novelty

The novelty of this work might be rather limited because the UAT2LLM has proved the uat of the transformer network.

Experiments

1) Numerical results are not convincing. The performance differences are very small such that it is not conclusive to say that the MRFNET is better than other methods.

2) Statistical tests need to be conducted to understand whether the performance differences are statistically significant or not.

3) Complexity analysis is absent. It is interesting to know the complexity of the proposed method compared to SOTA algorithms.

Conclusion

limitation of this method should be discussed in the conclusion section.

**Questions:**

1) I wonder the complexity of the proposed method.
2) I wonder the novelty of this paper provided that the UAT of various modules of neural networks have existed in the literature.

---

> ### Author Response · Authors · 2024-11-13
> **Official Comment by Authors**
>
> ### Weaknesses:
>
> >1. Motivation
>
> The research motivation of this paper is given in line 66 of the text: This paper aims to bridge the theoretical gap and practices in time series forecasting.
>
> We have provided the proof about the fact that neural network is a universal approximator in Section 3, Section4.1 and Section4.2. The proof approach in this paper is based on the methods used in UAT2LLM and UAT2CV.
>
> Here, we briefly outline the proof strategy presented in the paper. The basic operation in UAT involves matrix-vector multiplication of the form $\alpha_j \sigma\left(\mathbf{W}_j^{\mathrm{T}} \mathbf{x}+\theta_j\right)$, where $\mathbf{W}_j^{\mathrm{T}} \mathbf{x}$ can understood as various network modules, such as convolution. To establish the relationship between $\mathbf{W}_j^{\mathrm{T}} \mathbf{x}$ and fundamental operations like convolution, we introduce the matrix-vector method, as detailed in Section 3. Based on this approach, we represent 1D convolution as $\mathbf{W}_j^{\mathrm{T}} \mathbf{x}$ in Section 4.1. To demonstrate that the overall mathematical form of a multilayer network aligns with that of the UAT, we provide the corresponding mathematical formulation for a multilayer network in Section 4.2. Through this derivation, we  clarify that neural networks function as universal approximators.
>
> We are unclear about what the reviewer means by "no figures." If this refers to the performance bottleneck, Table 2 in the paper provides a substantial amount of data on this topic.
> > 2. Literature Survey
>
> Regarding the literature review, we are unable to cover all existing literature. We have made every effort to compare our work with the most popular and relevant methods. If the reviewer believes that any specific references should be included, please feel free to provide them to us.
> >3. Novelty
>
> The focus of this paper is on time series problems. While UAT2LLM has already demonstrated the UAT for transformer networks, it does not provide a proof for 1D convolution. In this paper, we provide that proof and discuss time series forecasting issues based on UAT.
>
> > Experiments
> > 1. Numerical results are not convincing. The performance differences are very small such that it is not conclusive to say that the MRFNET is better than other methods.
>
> First, it is a fact that MRFNET outperforms the current state-of-the-art models on most datasets. We believe this demonstrates that MRFNET is superior to other methods. As we discuss in our paper, current time-series forecasting networks are based on UAT, which leads them to encounter similar bottlenecks. However, our paper further explores ways to enhance performance in light of these limitations by leveraging specific data characteristics. As shown in Tables 3 and 4 in Section 5.4, we achieve significant improvements in the performance of current time-series forecasting models.
>
> > 2. Statistical tests need to be conducted to understand whether the performance differences are statistically significant or not.
>
> Could you please clarify what statistical tests the reviewers expect to be conducted? As far as we know, most current time series forecasting papers, such as DLinear and PatchTST, do not include any statistical tests. Therefore, our paper follows the writing tradition commonly adopted in time series forecasting research.
>
> > 3. Complexity analysis is absent. It is interesting to know the complexity of the proposed method compared to SOTA algorithms.
>
> We currently do not have a complexity analysis, but we will add this section later. Here, we briefly provide the algorithm's complexity for each layer as: T * T * C * K, where T, C, and K represent time, feature dimension size, and kernel size, respectively. However, we believe that a complexity analysis is not absolutely essential, as many time-series prediction papers, such as PatchTST, do not include one.
>
> Besides, due to different methods of complexity calculation, we believe that complexity metrics alone do not accurately reflect the true circumstances. FEDformer claims its time and space complexity to be $O(L) $. Since most models do not provide complexity metrics, we use the TimesNet paper as an example:
>
> - TimesNet: GPU Memory: 1585M, Running Time: 0.040s / iter
> - MRFNet: GPU Memory: 724M, Running Time: 0.0154s / iter
> - FEDformer: GPU Memory: 7111M, Running Time: 1.055s / iter
>
> **Remark:** Thanks for your constructive comments. Your consideration of improving the rating of our paper will be much appreciated!

---

> ### Author Response · Authors · 2024-11-13
> **Official Comment by Authors**
>
> ### Questions
>
> > 1. I wonder the complexity of the proposed method.
>
> We have given in Weaknesses.Experiments.3.
>
> > 2. I wonder the novelty of this paper provided that the UAT of various modules of neural networks have existed in the literature.
>
> This paper bridges the theoretical gap and practices in time series forecasting. Besides, the 1D convolution did not exist in the literature and we provide in this paper. Our contributions are as follows:
>
> We have demonstrated that most of the current deep-learning-based time series prediction models are specific implementations of the UAT, and explained why the current SOTA models converge to similar performance bottlenecks.
>
> We proposed the MRFNet model based on the characteristics of time series prediction, which balances both global and local temporal learning. Our model achieved SOTA performances across multiple datasets, and we also proved that this model is a specific implementation of UAT for time series prediction.
>
> We rearranged the data based on the inherent characteristics of time series data, which improved the predictive performance of the model largely.
>
> **Remark:** Thanks for your constructive comments. Your consideration of improving the rating of our paper will be much appreciated!

---

> ### Author Response · Authors · 2024-11-21
> **Sincerely looking forward to your feedback**
>
> Dear Reviewer T3gY,
>
> We hope our point-by-point response in the rebuttal has addressed your concerns. We are very much looking forward to your feedback during the discussion period. We would be more than happy to answer any further questions you may have.
>
> Best regards,
>
> The Authors

---

### Official Review · Reviewer_R7Yj · 2024-10-31

**Soundness:** 2
**Presentation:** 2
**Contribution:** 2
**Rating:** 5
**Confidence:** 2

**Summary:**

MRFNet is a novel multivariable time series prediction model, which is designed based on universal approximation theorem (UAT) and integrates linear, sparse matrix, convolution and Fourier transform modules. The model performs well on multiple popular data sets and achieves better prediction performance. The study also found that current state-of-the-art time series prediction models tend to have similar performance bottlenecks because they share the same mathematical foundation. MRFNet provides a new perspective to understand and improve time series forecasting by capturing both global and local information.

**Strengths:**

The MRFNet model combines multiple deep learning techniques, including linear, convolution, and Fourier transform, to accommodate the complexity of time series data. The design is based on the general approximation theorem, which ensures that the model can capture global and local information and improve the prediction accuracy.

**Weaknesses:**

MRFNet's module consolidation, while enhancing model functionality, is limited in innovation and relies primarily on a portfolio of existing technologies. This increases the complexity of the model.

**Questions:**

1. Capturing global and local features seems to be widely studied, so please identify the innovations in this article
[1] Dai, T., Wu, B., Liu, P., Li, N., Bao, J., Jiang, Y., & Xia, S. T. (2024). Periodicity decoupling framework for long-term series forecasting. In The Twelfth International Conference on Learning Representations.
2.MRFNet has limited performance improvements on some datasets. Do authors think this is mainly due to data characteristics or limitations of the model itself?

---

> ### Author Response · Authors · 2024-11-13
> **Official Comment by Authors**
>
> Weaknesses
>
> >Limited in innovation and relies primarily on a portfolio of existing technologies. This increases the complexity of the model.
>
> Regarding the novelty of our work, our primary contribution lies in designing the network from the theoretical perspective of UAT and applying this approach to analyze time series problems. To the best of our knowledge, no prior research has explored this direction.
>
> We would also like to emphasize that, although the proposed model may appear complex at a high level, it is actually built upon very basic operations, making implementation straightforward. For instance, compared to models like TimesNet and FEDformer, our model architecture is considerably simpler. While we have made our code open-source; however, I do not know weather I can share it now. If yes, I can provide anonymous link. We can provide some information comparing our model with other models to demonstrate that our network is not overly complex:
>
> TimesNet: GPU Memory: 1585M, Running Time: 0.040s / iter
> MRFNet: GPU Memory: 724M, Running Time: 0.0154s / iter
> FEDformer: GPU Memory: 7111M, Running Time: 1.055s / iter
>
> ### Questions
> >1.Capturing global and local features seems to be widely studied, so please identify the innovations in this article [1] Dai, T., Wu, B., Liu, P., Li, N., Bao, J., Jiang, Y., & Xia, S. T. (2024). Periodicity decoupling framework for long-term series forecasting. In The Twelfth International Conference on Learning Representations.
>
> Thank you for the reviewer’s suggestion; we will add a citation to this article in our next revision.
>
> > 2.MRFNet has limited performance improvements on some datasets. Do authors think this is mainly due to data characteristics or limitations of the model itself?
>
> We believe the primary issue lies with the data rather than the model itself. Based on UAT2LLM [1] and UAT2CV [2], as well as our discussions in Section 3, Section 4.1, and Section:4.2, we can conclude that the overall mathematical structure of most time series forecasting models aligns with UAT. This shared theoretical foundation means that they are all likely to encounter similar performance bottlenecks. However, in Section 5.4 of this paper, we address issues related to the data itself, leading to substantial performance improvements.
>
> **Remark:** Thanks for your constructive comments. Your consideration of improving the rating of our paper will be much appreciated!

---

> ### Author Response · Authors · 2024-11-21
> **Sincerely looking forward to your feedback**
>
> Dear Reviewer R7Yj,
>
> We hope our point-by-point response in the rebuttal has addressed your concerns. We are very much looking forward to your feedback during the discussion period. We would be more than happy to answer any further questions you may have.
>
> Best regards,
>
> The Authors

---

### Official Review · Reviewer_oZrE · 2024-11-02

**Soundness:** 1
**Presentation:** 3
**Contribution:** 2
**Rating:** 3
**Confidence:** 3

**Summary:**

## Summary of the Paper
The paper introduces MRFNet, a multivariate time-series forecasting model designed to capture patterns by integrating four components: linear, convolutional, sparse matrix, and Fourier transform blocks.

**Strengths:**

## Strong Points
1.	**Clear Presentation**
The paper is well-organized, making it easy to follow the transition from problem statement to proposed solution.
2.	**Introduction of UAT to Time-Series Forecasting**
The application of the Universal Approximation Theorem (UAT) in time-series forecasting adds an interesting theoretical perspective.

**Weaknesses:**

## Weak Points and Concerns
1.	**Insufficient Experiment Details and Missing State-of-the-Art References (Ref 1 and Ref 2 below)**
The experimental setup lacks details, such as the size of weight matrices, learning rates, training epochs, and batch sizes, which are essential for reproducibility. This missing information makes it difficult to replicate the experiments or understand how hyperparameter choices might affect the results. Additionally, comparisons with recent state-of-the-art methods are absent.
The improvement seems marginal.

Ref 1 (ICML 2024): SparseTSF: Modeling Long-term Time Series Forecasting with 1k Parameters
https://arxiv.org/pdf/2405.00946
Ref2 (ICLR 2024) :CARD: Channel Aligned Robust Blend Transformer for Time Series Forecasting
https://openreview.net/pdf?id=MJksrOhurE

2.	**Lack of Motivation for Selecting the Four Specific Blocks**
While MRFNet includes linear, convolutional, sparse matrix, and Fourier transform modules, there is no clear justification of using these four components over others.

3.	**Limited Theoretical Justification for UAT Application**
Although UAT is referenced as a motivating factor, the paper does not fully explain how each component in MRFNet contributes to leveraging UAT in time-series forecasting. This weakens the theoretical foundation.

**Questions:**

## Questions for Authors
1.	Can you provide more details on the experiment setup, specifically weight matrix sizes, learning rates, and training epochs?
2.	What motivated the selection of these four specific modules? Could alternative components potentially yield similar or improved results?
3.	How does MRFNet compare with recent state-of-the-art methods in the references?

---

> ### Author Response · Authors · 2024-11-14
>
> ### Weak Points and Concerns & Questions
>
> > 1. Insufficient Experiment Details and Missing State-of-the-Art References and comparisons with recent state-of-the-art methods are absent. The improvement seems marginal.
>
> Regarding experimental details, we have made our code open-source; however, I do not know weather I can share it now. If yes, I can provide anonymous link.
>
> Since I am unsure of your specific interests, I'll provide some basic parameters here. If you have additional requirements, we’d be happy to provide more details.
>
> - Learning rate: 0.0001
> - Epochs: 50 (with early stop strategy)
> - Batch size: 32
> - Input sequence length: 96 for ILI, 720 for all other data
> - Weight matrix sizes: 720*720
>
> On the matter of citing the following articles, we would first like to emphasize that there are numerous studies in the field of time series forecasting, and it is not feasible to reference all of them individually. We will, however, add citations to these two articles in future versions. Additionally, you may refer to our results in Tables 2 and 3, and compare them with the results of these articles. It is evident that our results outperform those presented in these two papers.
>
> As we mentioned, time series prediction has encountered a performance bottleneck, primarily because most of these networks are based on UAT theory, and our network architecture is no exception. Therefore, we have made only minor improvements by leveraging the specific characteristics of time series data. However, we further propose methods to address the bottleneck by utilizing the unique features of time series data. Please refer to Section 5.4 for details. We have made significant improvements in time series prediction by addressing these data issues, as shown in Tables 3 and 4.
>
> > 2. Lack of Motivation for Selecting the Four Specific Blocks:
>
> We provide the rationale for using these four components in Section MRFNET, specifically on line 320 of the paper. If you have any more question, please feel free to ask us.
>
> >3. Limited Theoretical Justification for UAT Application
>
> Firstly, UAT serves as the theoretical foundation of our paper, and we illustrate the relationship between UAT and these four components within the text.
>
> This relationship is established through the Matrix-Vector Method. The fundamental operation in UAT is a matrix-vector multiplication of the form $\alpha_j \sigma\left(\mathbf{W}_j^{\mathrm{T}} \mathbf{x}+\theta_j\right)$, where $\mathbf{W} _ j^{\mathrm{T}} \mathbf{x}$ can understood as various network modules, like convolution. Each of the four components discussed in the paper can be represented using this matrix-vector format, aligning with UAT’s core computation unit $\mathbf{W}_j^{\mathrm{T}} \mathbf{x}$ .
>
> We provide explanations of this relationship in Section 3 (line 124), Section 4.1 (line 199), and Section 5.1 (line 320).
>
> **Remark:** Thanks for your constructive comments. Your consideration of improving the rating of our paper will be much appreciated!

---

> ### Author Response · Authors · 2024-11-21
> **Sincerely looking forward to your feedback**
>
> Dear Reviewer  oZrE,
>
> We hope our point-by-point response in the rebuttal has addressed your concerns. We are very much looking forward to your feedback during the discussion period. We would be more than happy to answer any further questions you may have.
>
> Best regards,
>
> The Authors

---

### Official Review · Reviewer_fwWp · 2024-11-05

**Soundness:** 2
**Presentation:** 3
**Contribution:** 2
**Rating:** 3
**Confidence:** 3

**Summary:**

The paper introduces MRFNET — a new architecture for long sequence time series forecasting, using a mixture of linear, sparse matrix, convolutional, and Fourier blocks. They also express CNNs and transformers in matrix vector form, providing a more direct link to the universal approximation theorem.

**Strengths:**

While transformer based forecasting models have demonstrated large improvements in Long Sequence Time Series Forecasting (LSTF) tasks, improvements have largely been empirical in nature — with little formal analysis to understand the inner workings of the models.

**Weaknesses:**

However, I do have some fundamental concerns surrounding the motivations of the paper itself.

Firstly, while universal approximation is the cornerstone of function approximation and that building blocks (FFNs, CNNs, transformers) can be expressed in UAT form -- the inductive biases provided through architecture (and how they are aligned with data characteristics) can make a big impact on performance. This has been seen frequently outside of time series datasets (e.g. transformers for LLMs, CNNs for computer vision), and also in the way deep neural networks have evolved beyond simple MLPs (which are also universal function approximators) . As such, the conjecture that performance bottlenecks are due to limitations under UAT would require a lot more evidence than supplied in the paper.

Secondly, assuming this is the case, it is not immediately clear how/why the architecture allows MRFNET to overcome the performance limitations described. Furthermore, the building blocks described have all been utilised in other forecasting architectures, so it would also be important to describe why the current configuration is expected to improve performance.

**Questions:**

1. Is MRFNet a UAT based, and if not how is it different and how do differences result in performance improvements?
2. Are there any concrete examples of a purely cnn-based time series forecasting model bottle necking at the same performance as a similar transformer-based model? Would we expect the performance of a simple CNN/naive Transformer model to match PatchTST?
3. How significant are performance improvements over PatchTST (e.g. confidence intervals around improvements)?

---

> ### Author Response · Authors · 2024-11-14
> **Rebuttal by Authors**
>
> ### Weaknesses
>
> >W1: More evidence than supplied in the paper.
>
> Regarding the performance bottleneck associated with UAT and the role of network architecture in improving performance, we consider this to be a theoretical issue. This topic has already been addressed from a theoretical perspective in the works UAT2LLM [1] and UAT2CV [2]. Here, we briefly summarize the conclusions:
>
> The performance limitations of MLPs arise from their strict adherence to the UAT function. While UAT provides strong approximation capabilities, it cannot fully capture the complexity of the real world, which is inherently diverse and multi-faceted. One of the most effective strategies for improving performance through network architecture is the use of residual connections, which have been employed in both transformer and convolutional neural network (CNN) models. Residual connections are effective because, while their overall mathematical form aligns with UAT, their parameters are functions of the input, allowing them to adapt dynamically and enhance performance. We also show why residual-based networks are dynamic approximator in Section 4.2, because some bias terms are the functions of the input.
>
> This performance bottleneck is prevalent in both CV and time series forecasting, serving as real-world examples of this limitation. More example of bottleneck in time series prediction, you can refer to the Table 5 in this paper. Another cause of the bottleneck lies in the data itself, such as limited data availability or issues with data correlation. We discuss this further in the Section 5.4 (line 448), where we address these issues and achieve substantial performance improvements, as shown in Tables 3 and 4.
>
>
> [1] W. Wang and Q. Li, “Universal approximation theory: The basic theory for transformer-based large language models,” 2024. [Online]. Available: https://api.semanticscholar.org/CorpusID:271902755
> [2] W. Wang and Q. Li, “Universal approximation theory: The basic theory for deep learning-based computer vision models,” ArXiv, vol. abs/2407.17480, 2024. [Online]. Available: https://api.semanticscholar.org/CorpusID:271432349
>
> >W2: How/Why the architecture allows MRFNET to overcome the performance limitations described. Furthermore, the building blocks described have all been utilised in other forecasting architectures, so it would also be important to describe why the current configuration is expected to improve performance.
>
> In Section 5.1 of our paper, we provide an explanation for why the current configuration is effective. In essence, this effectiveness arises because the configuration enables the model to learn both global and local temporal and feature correlations, thus enhancing overall performance. To our knowledge, no prior work has used sparse matrices in the same way as described here, making this approach novel in our context.
>
> In addition, our paper does not claim that we overcome the time series prediction bottleneck through network design. This bottleneck is inherent to the UAT theory itself, and since our model also falls under UAT, it essentially only makes minor improvements for time series problems. However, our paper continues to address the bottleneck caused by another factor: data issues. We have made significant improvements in time series prediction by addressing these data issues, as shown in Tables 3 and 4.
>
> **Remark:** Thanks for your constructive comments. Your consideration of improving the rating of our paper will be much appreciated!

---

> ### Author Response · Authors · 2024-11-14
> **Rebuttal by Authors**
>
> ### Questions
> >Q1. Is MRFNet a UAT based, and if not how is it different and how do differences result in performance improvements?
>
> Firstly, MRFNet is based on the UAT. Our proof strategy aligns with the approaches in UAT2LLM[1] and UAT2CV[2], and the basic proof steps are detailed in lines [148-167] and Section: UNIVERSAL APPROXIMATION THEORY FOR TIME SERIES PREDICTION of the paper. To facilitate understanding for the reviewers, we elaborate here on how the proof is constructed:
>
> 1. Matrix-Vector multiplication is the core element in UAT: The fundamental operation in UAT is matrix-vector multiplication of the form $\alpha_j \sigma\left(\mathbf{W}_j^{\mathrm{T}} \mathbf{x}+\theta_j\right)$, where $\mathbf{W}_j^{\mathrm{T}} \mathbf{x}$ can represent various network modules. In temporal models, this is typically represented as linear or 1D convolution operations. Linear transformations have been shown to fit within the  $\mathbf{W}_j^{\mathrm{T}} \mathbf{x}$  in UAT2LLM; So in this paper, we primarily focus on demonstrating that 1D convolution can similarly be expressed in matrix-vector form. This can be further referenced in the Section: Universal Approximation Theory for Time Series Prediction.
>
> 2. Representing temporal prediction operations in matrix-vector form: Once we express the fundamental operation units of temporal prediction in matrix-vector form (lines [147-167] and [200-255]), we then extend this to multi-layer networks. We observe that the resulting mathematical structure aligns closely with the form of UAT, which can be found in Section: 4.2 THE UAT FORMAT OF DEEP LEARNING NETWORKS IN TIME SERIES PREDICTION(lines [250-308]).
>
> 3. Example of a Two-Layer MRFNet in UAT Form for Clarity: To aid understanding, we present here the UAT form corresponding to a two-layer MRFNet, as additional layers can complicate the interpretation. Should reviewers need a complete proof, we are prepared to provide it in full.
>
> $$
> \mathbf{x} _ {2}'=(\mathbf{W} _ {2,1}'\mathbf{x} _ 0'+\mathbf{b} _ {2,1}')+\mathbf{W} _ {2,1}'\sigma
> (\mathbf{W} _ {1,2}'\mathbf{x} _ 0'+\mathbf{b} _ {1,2}')
> +\mathbf{W} _ {2,1}'\sigma
> (\mathbf{W} _ {1,3}'\mathbf{x} _ 0'+\mathbf{b} _ {1,3}')+\mathbf{W}  _  {2,1}'\sigma
> (\mathbf{W} _ {1,4}'\mathbf{x} _ 0'+\mathbf{b} _ {1,4}')
> +\sigma
> (\mathbf{W} _ {2,2}'\mathbf{x} _ 0'+\mathbf{b} _ {2,2}')+\sigma(\mathbf{W}  _ {2,3}'\mathbf{x} _ 0'+\mathbf{b} _ {2,3}')
> +\sigma(\mathbf{W} _ {2,4}'\mathbf{x} _ 0'+\mathbf{b} _ {2,4}')
> $$
>
> [1] W. Wang and Q. Li, “Universal approximation theory: The basic theory for transformer-based large language models,” 2024. [Online]. Available: https://api.semanticscholar.org/CorpusID:271902755
>
> [2] W. Wang and Q. Li, “Universal approximation theory: The basic theory for deep learning-based computer vision models,” ArXiv, vol. abs/2407.17480, 2024. [Online]. Available: https://api.semanticscholar.org/CorpusID:271432349
>
>
>
> > Q2. Are there any concrete examples of a purely cnn-based time series forecasting model bottle necking at the same performance as a similar transformer-based model? Would we expect the performance of a simple CNN/naive Transformer model to match PatchTST?
>
> Based on Table 2 in line 385 and Table 5 in line 1055, it is clear that nearly all current models are approaching the same performance bottleneck. These tables provide a comprehensive comparison of the performance of most SOTA models. By "performance bottleneck," we are not only referring to purely CNN-based time series forecasting models and transformer-based models but also to various improved versions of these architectures.
>
> A simple CNN cannot match the performance of PatchTST because the mathematical form of a simple CNN is essentially the same as that of the UAT which can refer to UAT2CV[1]. However, real-world data is diverse, which necessitates the use of residual connections. Residual structures enable the parameters in the UAT to dynamically approximate based on the input, as explained in Section 4.2. This is also why most current models, including Transformers, employ residual connections. PatchTST itself is a variant of the Transformer. Therefore, the most crucial factor is the residual connection. Beyond that, we can design the network further based on the specific characteristics of time-series data.
>
> [1] W. Wang and Q. Li, “Universal approximation theory: The basic theory for deep learning-based computer vision models,” ArXiv, vol. abs/2407.17480, 2024. [Online]. Available: https://api.semanticscholar.org/CorpusID:271432349
>
>
> >Q3. How significant are performance improvements over PatchTST (e.g. confidence intervals around improvements)?
>
> Our performance improvements are shown in Table 2. Since most time series forecasting models do not discuss confidence intervals, we have also omitted this aspect. But, we achieved significant performance enhancements using a feature decomposition approach, as presented in Tables 3 and 4.

---

> ### Author Response · Authors · 2024-11-21
> **Sincerely looking forward to your feedback**
>
> Dear Reviewer  fwWp,
>
> We hope our point-by-point response in the rebuttal has addressed your concerns. We are very much looking forward to your feedback during the discussion period. We would be more than happy to answer any further questions you may have.
>
> Best regards,
>
> The Authors

---

> ### Comment · Reviewer_fwWp · 2024-11-26
>
> Thank your detailed response. After going through the reply and comments from other reviewers, I do agree with other fellow reviewers on the limitations regarding novelty and marginal performance improvements shown in the paper - leading me to maintain my current score.
>
> From the below - the main source of performance improvements appears to be from customising training to nuances in specific datasets. Namely 1) leveraging walkforward validation for ETT* data to overcome non-stationarity (not periodicity I think, as seasonality can be learnt from data), and 2) what seems like feature selection for ILI, with other similar extensions omitted on the rest (as MRFNET also bottlenecks to UAT limit). While I do agree that leveraging domain knowledge is a good way to get dataset-specific improvements, success would likely vary on a case-by-case basis and tied to specific applications.

---

### Author Response · Authors · 2024-11-15
**Rebuttal by Authors**

General reply
We deeply appreciate the insightful and constructive comments from the reviewers, which are helpful in improving our paper. We are pleased that all the reviewers recognized the contribution in theory.

We have summarized a detailed reply to several common questions and addressed other concerns in each individual rebuttal.

>1.The performance improvement of our model compared to other models is relatively small.

As mentioned in our paper, most current time series prediction models fall under the category of UAT functions, and our model is no exception. Therefore, given the same theoretical foundation, their performance tends to converge towards the same bottleneck. Our paper addresses the characteristics of time series data in the network design and achieves SOAT standards on multiple datasets, which we believe demonstrates the effectiveness of our model.

Additionally, we propose a method to address the performance bottleneck by focusing on the data itself in Section 5.4. Based on this approach, we have significantly improved model performance, achieving results that far exceed current SOTA results.

In response to the reviewers' comments, we have made the following modifications to the manuscript (You can find the pdf in Supplementary Material):

1. **Addressing the concern regarding MRFNet's performance improvement over other networks:**
   In Section 5.3, we have explicitly clarified that since our network falls within the UAT function framework, its performance improvements, even with designs tailored for temporal problems, are inherently limited to modest enhancements.
   However, we conducted more in-depth data processing and training optimizations tailored to the unique properties of temporal data. The results in Tables 4 and 5 demonstrate that these efforts significantly improved MRFNet’s predictive performance.

2. **Incorporating references suggested by reviewers oZrE and R7Yj:**
   We have added citations to the literature provided by these reviewers in Section 2.

3. **Adding complexity analysis as requested by reviewer T3gY:**
   A detailed complexity analysis has been included in Section 5.1.

4. **Adding limitations to the conclusion section:**
    Following reviewer T3gY's suggestion, we have added a discussion of the model’s limitations in the conclusion section.

---

> ### Author Response · Authors · 2024-11-21
> **Look forward to further discussions before the end of the discussion period**
>
> Dear Reviewers:
>
> As the author-reviewer discussion period will end soon, we would appreciate it if you could kindly review our responses at your earliest convenience. If there are any further questions or comments, we will do our best to address them before the discussion period ends.
>
> Thank you very much for your time and efforts!
>
> Sincerely,
>
> The Authors

---

> ### Author Response · Authors · 2024-11-25
> **Look forward to further discussions before the end of the discussion period**
>
> Dear Reviewers:
>
> As the author-reviewer discussion period will end soon, we would appreciate it if you could kindly review our responses at your earliest convenience. If there are any further questions or comments, we will do our best to address them before the discussion period ends.
>
> Thank you very much for your time and efforts!
>
> Sincerely,
>
> The Authors

---

### Note · Authors · 2024-11-28

I have read and agree with the venue's withdrawal policy on behalf of myself and my co-authors.